# Intermittent rate coding and cue-specific ensembles support working memory

Matthew F. Panichello[1,2 ✉], Donatas Jonikaitis[1,2], Yu Jin Oh[1], Shude Zhu[1], Ethan B. Trepka[1] & Tirin Moore[1 ✉]

Persistent, memorandum-specific neuronal spiking activity has long been hypothesized to underlie working memory[1,2]. However, emerging evidence suggests a potential role for 'activity-silent' synaptic mechanisms[3–5]. This issue remains controversial because evidence for either view has largely relied either on datasets that fail to capture single-trial population dynamics or on indirect measures of neuronal spiking. We addressed this controversy by examining the dynamics of mnemonic information on single trials obtained from large, local populations of lateral prefrontal neurons recorded simultaneously in monkeys performing a working memory task. Here we show that mnemonic information does not persist in the spiking activity of neuronal populations during memory delays, but instead alternates between coordinated 'On' and 'Off' states. At the level of single neurons, Off states are driven by both a loss of selectivity for memoranda and a return of firing rates to spontaneous levels. Further exploiting the large-scale recordings used here, we show that mnemonic information is available in the patterns of functional connections among neuronal ensembles during Off states. Our results suggest that intermittent periods of memorandum-specific spiking coexist with synaptic mechanisms to support working memory.

Working memory allows us to retain and manipulate information on short timescales, and it is central to complex cognitive processing and adaptive behaviour[6,7]. Foundational work in the 1970s showed that working memory is associated with sustained neuronal spiking activity in primate prefrontal cortex[8]. Subsequent studies demonstrated that the persistent spiking of many neurons is specific to a remembered cue[9–11]. Persistent activity has been observed during both spatial and feature-based working memory tasks[12–14], as well as within many cortical and subcortical brain structures[15–20]. In addition to non-human primates, it has also been observed in multiple animal models[21] as well as in humans[22,23]. Combined, this evidence has established persistent spiking as the dominant model of working memory[2].

In spite of the predominance of the persistent-spiking model of working memory, an alternative class of models has received increased attention in recent years. This class of models proposes that working memory is supported by 'activity-silent', synaptic mechanisms rather than persistent activity[4,5,24]. Specifically, information held in working memory is stored by the pattern of short-term plastic changes initiated by a particular memory cue. Proof-of-principle simulations demonstrate that short-term plasticity (STP) can maintain information in the absence of persistent spiking[4,24]. Evidence of such latent traces has been reported using a variety of methods[25–29]. For example, STP, as inferred from functional connectivity, has been shown to correlate with cross-trial serial biases[25] and the maintenance of tasks sets[30] during working memory. However, it remains unknown whether cue-specific synaptic mechanisms operate during canonical working memory delays, when the maintenance of memorandum information is most critical.

Nevertheless, synaptic models of working memory can potentially address key shortcomings of the persistent-spiking model. For one, persistent activity has been reported to be modest, or even absent, in some cases[27–29,31], and to vary with task demands[26,32]. Second, and more importantly, delay-period activity can be highly variable on single trials[33], prompting some to question the utility of persistent spiking as a reliable mechanism for memory maintenance[3–5]. In addition, the high-gamma component of prefrontal local field potentials appears bursty, rather than persistent, during memory delays, suggesting that population spiking may be similarly irregular[34,35]. In principle, a synaptic mechanism could eliminate, or at least minimize, disruptions in memory maintenance due to spiking irregularities. However, the relative contributions of spiking and synaptic mechanisms to working memory remain largely unresolved.

To address the above questions, we studied the activity of neurons within the lateral prefrontal cortex (areas 8 and 9/46) in three monkeys (A, H and J; Methods). The monkeys were trained to perform one or two variants of a spatial working memory task (Fig. 1a). In both variants, the monkey was first presented with a brief (50 ms) spatial cue at one of eight possible locations while fixating a central spot. Following the cue, the monkey maintained fixation during a memory delay (1,400–1,600 ms). In one task (match-to-sample, MTS)[36], two targets appeared after the delay and the monkey was rewarded for making an eye movement to the target appearing at the previously cued location. In the

[1]Department of Neurobiology and Howard Hughes Medical Institute, Stanford University, Stanford, CA, USA. [2]These authors contributed equally: Matthew F. Panichello, Donatas Jonikaitis. ✉e-mail: mfp2@stanford.edu; tirin@stanford.edu

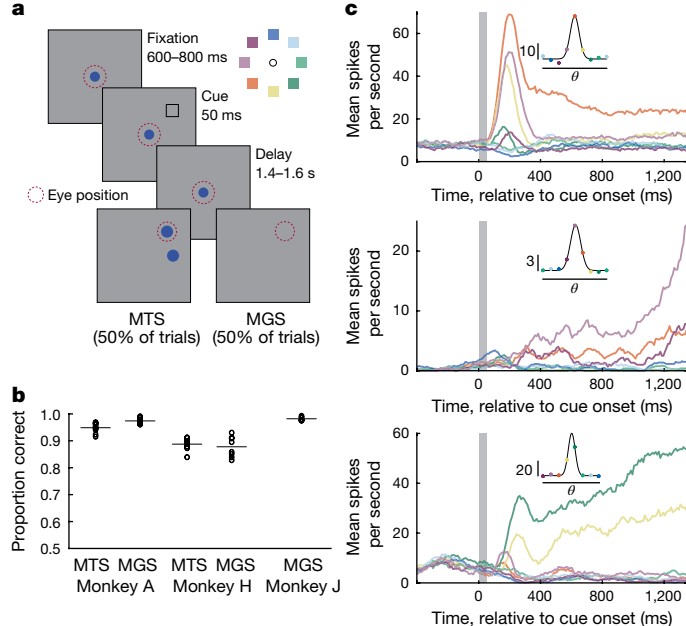

**Fig. 1 | Persistent, trial-averaged neuronal responses during spatial working memory. a**, Delayed MTS and MGS tasks. On each trial, the animal was presented with a cue at one of eight possible locations (inset). Following a memory delay period, the animal received fluid reward for making an eye movement to the previously cued location. **b**, Proportion correct for MTS and MGS tasks. Circles denote individual sessions, lines represent mean across $n = 25$ sessions. **c**, Trial-averaged peristimulus time histograms for three example prefrontal neurons showing canonical persistent activity during the memory delay period. Colours denote different cue locations. Insets show cue-location ($\theta$) tuning functions for each neuron during the memory delay, with scale bar (spikes per second) and gaussian fit (black trace).

second task (memory-guided saccade, MGS)[9], no targets appeared after the delay and the monkey was rewarded for making an eye movement to the previously cued (blank) location. All three monkeys achieved excellent performance (Fig. 1b). Trials of both task type were randomly interleaved and were pooled for subsequent analyses.

## High-density recordings from prefrontal cortex

As expected from previous studies[9,10], we observed a substantial proportion of prefrontal neurons with cue-specific memory delay activity (mean, 44% per session). Trial-averaged responses of these neurons suggest that their firing rates are sufficient to encode the remembered cue during the memory delay (Fig. 1c). However, trial averaging can obscure the high variability of spiking activity on single trials[3]. Thus, for individual neurons, cue information may be unreliable at times during the delay. Nonetheless, it could be that lapses in cue information for some neurons in the population are compensated by the continued activity of other neurons encoding the same memorandum. Alternatively, these lapses could be coordinated such that cue information fails to persist throughout the memory delay across the entire neuronal population. To distinguish between these possibilities, it is crucial to simultaneously measure the activity of large populations of neurons and to examine their activity on single trials, to evaluate the contribution of persistent spiking to working memory.

In the past few years, high-density, silicon probes, most notably Neuropixels probes (IMEC, Inc.), have revolutionized large-scale electrophysiological recordings in the mouse brain[37,38]. More recently, these probes were adapted for use in non-human primates[39]. We used these probes to obtain recordings from large, dense populations of prefrontal neurons in monkeys performing the spatial working memory

tasks (Fig. 2a). Our Neuropixels recordings ($n = 25$ sessions) typically yielded hundreds of single and multi-units in each session (mean, $329 \pm 46$; $n = 8,225$ total; Methods). In addition to memory delay neurons, these recordings allowed us to capture the spatial distribution of multiple functional classes of neurons (Extended Data Fig. 1). For example, neurons selective to multiple task components (for example, visuomotor neurons) tended to be more closely spaced than those selective to only one.

Most importantly, these recordings allowed us to quantify the information collectively conveyed by local populations of neurons about the remembered cue location. To do this, we used a leave-one-out, binary classification procedure. For each trial and time point within a session, we trained logistic regression models to discriminate the test location from its opposite location across the trial duration (Fig. 2b). For these and subsequent results, analyses of the memory delay were confined to the period from 500 to 1,400 ms following cue appearance, to avoid the influence of visually evoked responses (Methods). Across recording sessions, mean classification accuracy was significantly above chance throughout the memory delay (all $P < 0.001$, sign-rank) (Fig. 2c). Moreover, for each individual session, mean classification across the memory delay exceeded chance performance (all $P < 0.001$, sign-rank), with accuracies ranging from 59 to 89%. Last, classification accuracy was similar across cue locations (range, 67–74%, all $P < 0.001$, sign-rank) (Fig. 2d).

## Stability of cue information in firing rates

The persistence of cue information in the averaged classification accuracy during the memory delay, however robust, may nonetheless belie memory dynamics occurring on single trials. In particular, any lack of persistence on single trials could be obscured in the trial-averaged accuracy. To investigate this, we adapted techniques recently used to study value coding[40] to examine the single-trial dynamics of cue information during working memory. Specifically, we analysed the confidence of the classifier described above—the posterior probability assigned to the correct class at test, which provides a time-resolved index of the amount of cue information in population spiking during each trial (Methods). In each monkey, classifier confidence correlated with reaction time on correct trials (Extended Data Fig. 2a). Clearly, if cue information persists on single trials, confidence values should remain stably above chance (0.5) throughout the memory delay.

On the contrary, we found that confidence failed to persist through the memory delay on single trials. Instead, lapses in classifier confidence were evident throughout the memory delay and across trials within each recording session (Fig. 3a, Extended Data Fig. 3). At the start of each trial, confidence was consistently high during the visual response to the cue. However, following the disappearance of the cue, confidence often returned to chance multiple times during the memory delay. During single trials, periods of high confidence were interrupted by sharp transitions to low confidence (Fig. 3b). Lapses in confidence were not associated with microsaccades (Extended Data Fig. 4). Furthermore, these transitions between high and low confidence did not appear aligned across trials (Fig. 3a,b, Extended Data Fig. 3).

Given the apparent fluctuations between high and low confidence, we next sought to determine whether single-trial confidence was best described by one or two means. Our null hypothesis was that fluctuations in confidence reflect random perturbations around a single mean. We formalized this by fitting a single beta distribution, which is used to model the behaviour of random variables on the interval [0, 1], to the histogram of confidence values from each session (Methods). The alternative, two-mean model describes confidence using a mixture of two beta distributions, reflecting two discrete states. Indeed, we found that the cross-validated, two-state model outperformed the single-state model in 19 of 25 recording sessions; this was due to the inability

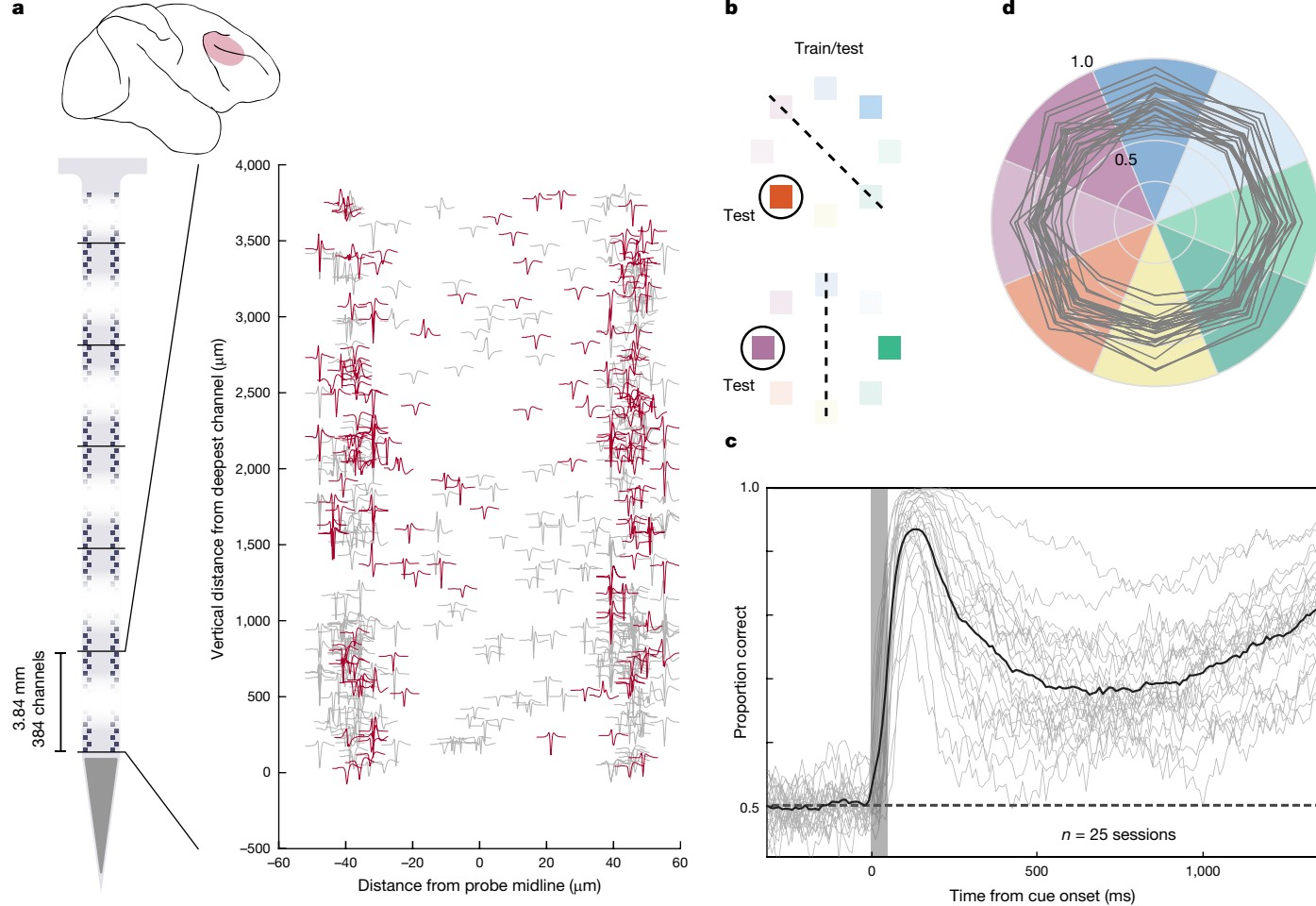

**Fig. 2 | High-density neuronal recordings from lateral prefrontal cortex.**
**a**, Top, location of recordings in lateral prefrontal cortex. Bottom left, schematic of the Neuropixels NHP probe, highlighting the contiguous block of 384 active channels near the probe tip. Bottom right, spike waveform templates for 480 single and multi-units extracted from a single example recording session in monkey A, shown at their measured location on the probe surface. Units plotted in red showed selectivity for cue location during the delay period.

**b**, Leave-one-trial-out training procedure. For each trial and time point, a classifier was trained on the remainder of trials to discriminate the same cue location as the test trial from the opposite cue location. **c**, Mean classification accuracies (proportion of trials correct) for cue location from $n = 25$ sessions, relative to cue onset. Traces indicate individual sessions. **d**, Mean proportion correct, averaged across the memory delay (+500 to +1,400 ms relative to cue onset) by cue location. Grey traces show individual sessions.

of the single-state model to capture the broad distribution of confidence values (Extended Data Fig. 5). This asymmetry in model performance was significant across recording sessions ($\chi^2(1) = 7$, $P = 0.009$). Similar results were obtained when using mixtures of Gaussians (Methods).

## Coordinated, intermittent rate-coding of memoranda

Having identified evidence of two discrete states, we next sought to label them on individual trials. We repeated the above classification procedure 50 times, shuffling condition labels on each iteration, to obtain a null distribution of confidence values for each trial (Fig. 4a and Methods). Across the cue and delay epochs, we labelled contiguous time points in which confidence was significantly greater than the null as 'On' states and labelled contiguous non-significant time points ($P > 0.20$) as 'Off' states. During each trial, we observed a mean of $2.35 \pm 0.04$ (median, 2.0) On states and a mean of $3.74 \pm 0.04$ (median, 4.0) Off states from the cue period until the end of the memory delay (Fig. 4b). Periods in which confidence was significantly below the null (confidently incorrect) were rare (mean, $0.11 \pm 0.01$ per trial). The mean duration of On states was $192.4 \pm 2.1$ ms (median, 150), and the mean duration of Off states was $146.1 \pm 1.7$ ms (median, 100)

(Fig. 4c). State (On or Off) at the time of the go cue predicted behavioural performance and reaction times in all three animals (Extended Data Fig. 2b,c). Importantly, classifiers trained only on Off states and tested on held-out Off states did not perform reliably above chance (Extended Data Fig. 6a), suggesting that Off states reflect time periods with no reliable information, and not a less frequent, second coding scheme that our original classifier failed to capture. In addition, we were not able to identify any predictive relationship between the phase of the local field potential along a range of frequency bands and On and Off states (Extended Data Fig. 6), suggesting that these fluctuations in confidence are distinct from the rhythmic sampling of attention described previously[41]. Finally, estimates of background noise[42] did not differ between On and Off states, indicating that they were not associated with fluctuations in recording quality (Extended Data Fig. 6c).

Having labelled On and Off states in this way, we next assessed their explanatory power and examined how they were reflected in the activity of individual neurons during the memory delay using cross-validation. To do this, we used half of the neurons recorded during each session to label states as On and Off and then examined the activity of neurons in the remaining (held-out) half of the population from the same sessions. We then repeated this process, switching training and test labels,

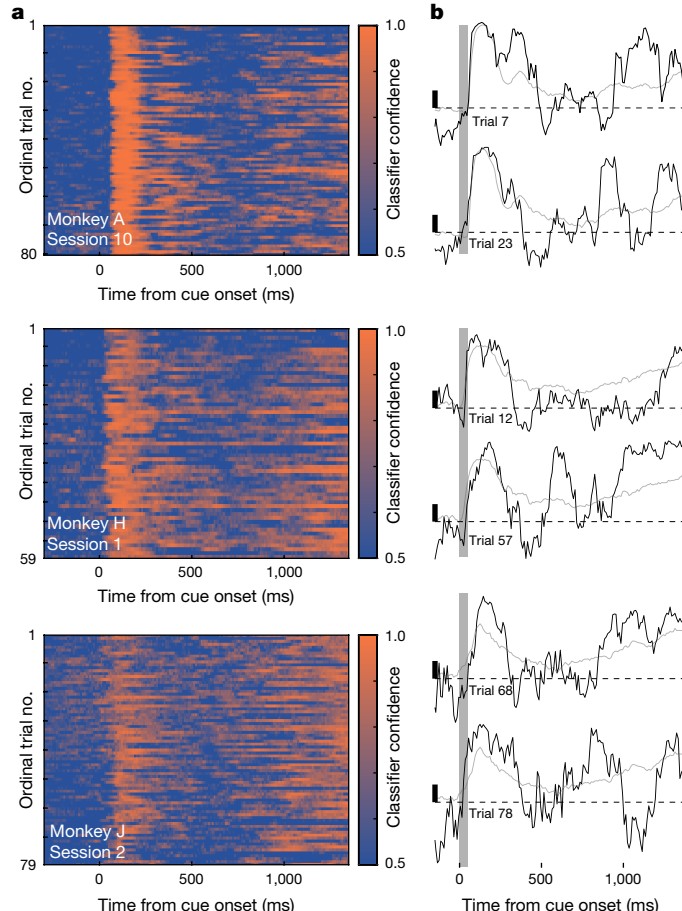

**Fig. 3 | Single-trial dynamics of memory signals in population spiking activity. a**, Single-trial classifier confidence, relative to cue onset, for all trials from the 'preferred' cue condition for three sessions. **b**, Confidence (black traces) for six example trials drawn from the three sessions in **a**. Grey traces show trial-averaged confidence values. Black scale bars denote a 0.10 increment in confidence; dashed lines denote $y = 0.50$.

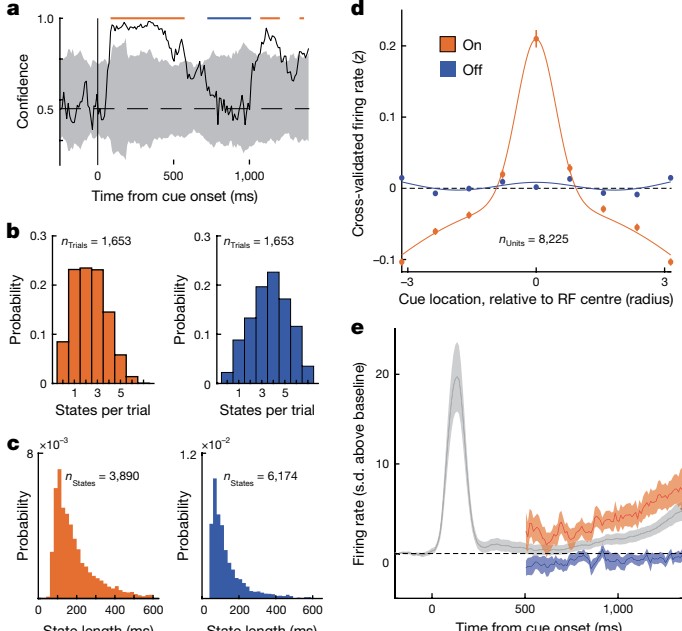

**Fig. 4 | Coordinated changes in memory selectivity and firing rates during On and Off states. a**, Example trial illustrating the labelling of On and Off states. **b,c**, Histograms of the number of On (orange) and Off (blue) states per trial (**b**, $n = 1,653$ trials) and state durations for $n = 3,890$ On and $n = 6,174$ Off states (**c**). **d**, Memory tuning functions for $n = 8,225$ held-out units during On and Off states. Tuning functions show the mean normalized firing rate during the memory delay ($z$-scored across trials) for held-out units, relative to each unit's preferred cue location. RF, receptive field. **e**, Mean normalized population firing rate (s.d. above spontaneous levels) across all $n = 25$ sessions for held-out units, relative to cue onset. Averages are plotted for all data points (grey), and also separately for firing rates extracted from On and Off states during the memory delay. **d,e**, Error bars denote mean ± s.e.m.

basic firing-rate properties of individual neurons. Furthermore, these observations show that transitions between On and Off states during the memory delay are coordinated across neurons within the local population.

## Cue-specific neuronal ensembles during memory delay

The fact that cue information carried by neuronal firing rates is periodically lost during memory delay suggests that persistent activity may not be sufficient to support working memory. Therefore, we next considered evidence in favour of synaptic models. These models propose that, rather than persistent activity, working memory is instead represented by cue-specific networks of neurons[4,5,24] (Fig. 5a)—that is, cue-specific neuronal ensembles should be a signature of working memory. Thus, we next looked for cue-specific cell assemblies during the memory delay. As in our analyses of firing-rate dynamics, we leveraged the Neuropixels recordings to measure functional connections among the very large numbers of simultaneously recorded neuronal pairs (mean, 52,314 ± 11,252 pairs per session). Specifically, we examined neuronal cross-correlations to assess their dependence on working memory.

To do this, we computed the cross-correlogram (CCG) between all neuronal pairs in each experimental session and for each cue condition based on activity during the memory delay. CCGs were computed, thresholded and firing rate normalized using established methods[45,46] (Fig. 5b, Extended Data Fig. 8 and Methods). We focused on CCGs with peaks at low latency (below 10 ms (refs. 45,47)), non-zero time lags, which are those most consistent with synaptic connections[48].

allowing an unbiased analysis of all 8,225 units (Methods). Indeed, we found that activity differed markedly between On and Off states in two ways. First, spatial tuning during the memory delay, a hallmark of spatial working memory[43,44], depended heavily on state. During On states, held-out neurons were strongly tuned to the location of the remembered cue (Fig. 4d and Extended Data Fig. 7). By contrast, during Off states, spatial tuning was virtually eliminated (Fig. 4d and Extended Data Fig. 7). Accordingly, there was a significant interaction between cue location and state (On or Off) on firing rates (two-way repeated measures analysis of variance (ANOVA), $P < 0.001$). Whereas cue location explained an average of 8.2% of variance in firing rate during On states, it explained only 0.7% during Off states, a 12-fold decrease. Thus, Off states were associated with a pronounced loss of spatial tuning at the level of individual neurons.

Second, average firing rates during the memory delay also depended on state. During On states, held-out neurons exhibited firing rates above spontaneous levels (mean 5.8 ± 0.4 Hz precue baseline) during the memory delay ($P < 0.001$, sign-rank; Fig. 4e). By contrast, during Off states, firing rates were statistically indistinguishable from spontaneous levels ($P = 0.192$) and significantly lower than On states ($P = 0.001$). Thus, Off states were associated not only with a loss of spatial selectivity but also with a collapse of firing rates to spontaneous levels. Together, these results show how transitions between On and Off states, derived from confidence values, reflect changes in

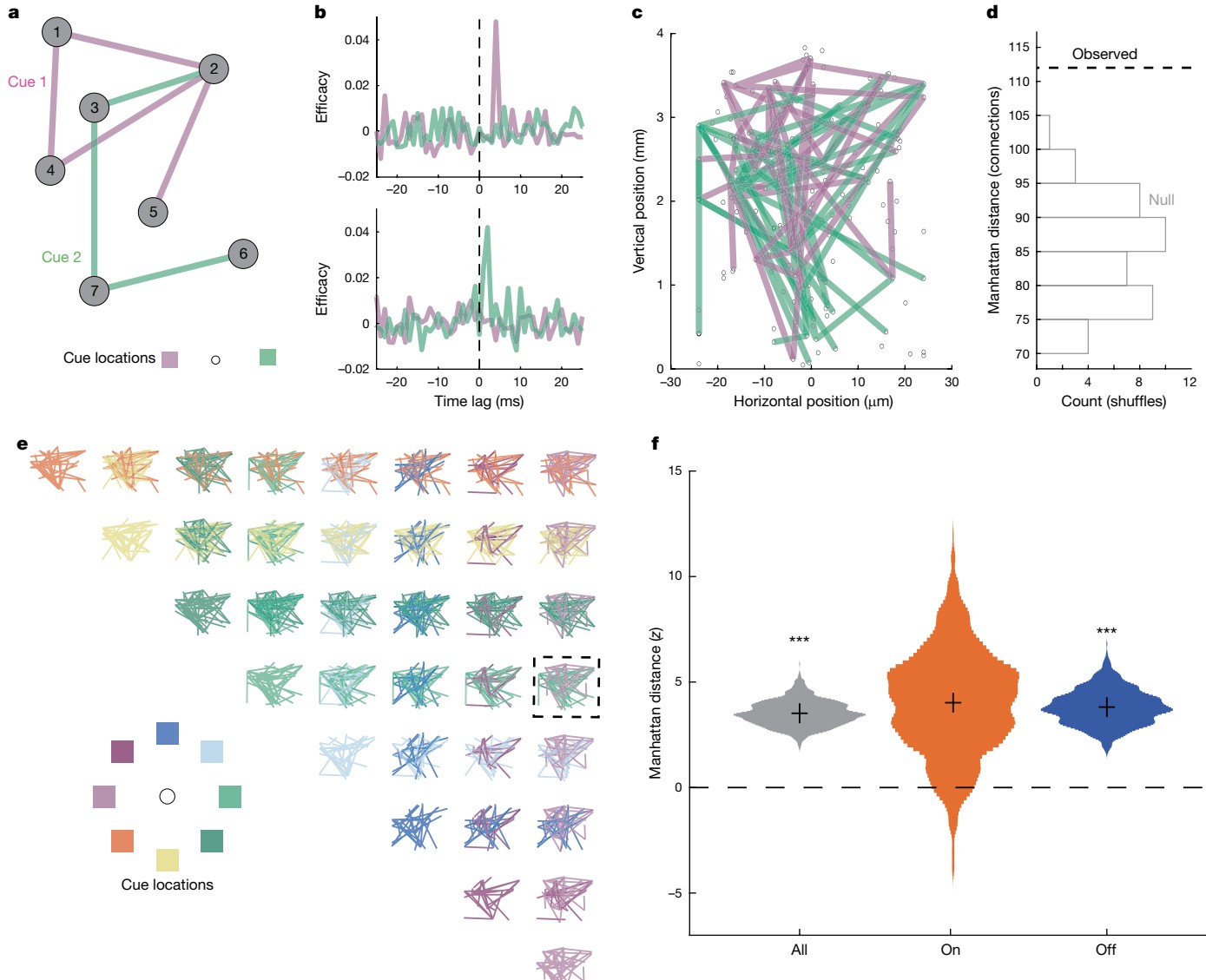

**Fig. 5 | Cue-specific neuronal ensembles during memory delay. a**, Cartoon depicting the synaptic model of working memory. In the absence of cue-selective firing rates, information persists in the cue-specific patterns of potentiated connections (pink and green lines) among neurons (grey nodes). **b**, Examples of pairwise CCGs computed during memory delay following two different cues. **c**, CCG-derived connectivity maps during memory delay for two different cue locations measured in one session. Lines are drawn between neuronal pairs exhibiting significant CCGs. **d**, Differences (dashed line) between the two connectivity maps, quantified as Manhattan distance:

the sum of cue-specific connections. Comparison of this metric to a null distribution derived from condition-shuffled data (grey-outlined bars) yielded a z-score (in this case, $z = 2.85$). **e**, Pairwise comparisons of connectivity maps across all cue locations. Dashed box shows comparison depicted in **c. f**, Mean normalized Manhattan distance (black crosses), using data from all $n = 25$ sessions during the entire memory delay (grey, $P = 1.2 \times 10^{-5}$), during On states only (orange, $P = 0.129$) and during Off states only (blue, $P = 2.5 \times 10^{-5}$). Violin plots show bootstrap across sessions. ***$P < 0.001$, two-sided sign-rank.

In our recordings, we observed CCGs with low-latency, suprathreshold peaks ('significant CCGs') of this type in $1.36 \pm 0.22\%$ of neuronal pairs, which totalled 12,740 significant CCGs across all sessions. Consistent with previous work[46], the probability of observing a significant CCG decreased with the distance between neurons ($r(39) = -0.42$, $P = 0.007$). Significant CCGs were found at all penetration sites. Next, we compared the pattern of CCGs across cue conditions. Figure 5c shows an example of a comparison of significant CCGs computed for two cue conditions (0 and 180°) in one recording session. In this example session, the two conditions exhibited highly dissimilar ensembles of functionally connected neurons. To quantify the dissimilarity, we counted the number of condition-unique CCGs, yielding the Manhattan distance, which could then be compared with a null distribution derived from condition-shuffled data (Fig. 5d). We then

repeated this procedure for all possible pairs of conditions in each session (Fig. 5e). Across sessions, mean Manhattan distance was significantly greater than that predicted by chance ($P < 0.001$, sign-rank; Fig. 5f and Extended Data Fig. 7). This effect remained significant when confined to a comparison of firing-rate-matched conditions ($P < 0.001$, sign-rank; Extended Data Fig. 9 and Methods). Thus, the ensemble of functionally connected neurons significantly depended on the remembered cue.

If the observed cue-specific ensembles support working memory, these should be evident when cue information in firing rates is absent. To test this, we repeated the above analysis separately for On and Off states. Notably, Manhattan distance was not significantly greater than chance during On states ($P = 0.129$, sign-rank; Fig. 5f and Extended Data Fig. 7). However, during Off states, when spatial tuning was

virtually absent and firing rates transitioned to spontaneous levels (Fig. 4d,e), Manhattan distance was significantly greater than chance ($P < 0.001$, sign-rank). This pattern of effects remained when confined to comparisons of firing-rate-matched conditions ($P = 0.062$ and 0.001 for On and Off states, respectively, sign-rank; Extended Data Fig. 9 and Methods). Thus, even in the absence of persistent memory delay activity, memoranda information was reflected in the cue-specific ensembles of functionally connected neurons.

How might spiking and activity-silent coding work together to support memory? Synaptic models of working memory propose that evoked responses to a memory cue potentiate synapses between cue-selective neurons through STP (Fig. 6a). During the subsequent memory delay, this evoked response relaxes to spontaneous levels. However, the cue-specific pattern of potentiated synapses remains. Consequently, cue-specific elevations in firing rate may nonetheless re-emerge owing to non-specific fluctuations in extrinsic or intrinsic activity. Thus, across a memory delay, multiple transitions between spiking and activity-silent modes (On and Off) may occur[4,24].

This account makes two testable predictions. First, it predicts a stable memory code—that is, despite the interruption of Off states, the same pattern of cue-selective spiking activity should support memory during On states throughout the memory delay. Indeed, similar to previous studies[49], we found that the trial-averaged performance of classifiers trained on cue-evoked responses successfully generalized across the entire memory delay (Fig. 6b). Demixed principle component analysis[50] and analysis of single neurons also supported a stable memory code (Extended Data Fig. 10). Second, there should be a correspondence between spiking responses evoked by the cue and functional connectivity observed during memory delay. Thus, two neurons exhibiting evoked responses to a given cue should tend to be functionally connected during memory. In our data, when considering any single cue location, 1.47 ± 0.4% of neuronal pairs responded preferentially to that cue location shortly after cue onset (average percentage across the eight possible cue locations). In addition, an average of 1.4 ± 0.2% of neuronal pairs exhibited significant CCGs to that cue location during memory delay (Fig. 6c). If these proportions are independent, their conjunction should occur at a rate equal to their product. On the contrary, we found that pairs showing spiking responses and significant CCGs to the same cue were observed 2.5 times more than expected ($P < 0.001$, sign-rank; Fig. 6d). Thus, neurons that responded jointly during the evoked response were more likely to be functionally connected during memory delay, consistent with synaptic models. Overall, 55% of neurons showed a cue-specific evoked response, 81% were involved in a cue-specific functional connection and 10.4% were involved in such jointly selective and connected pairs.

## Discussion

These results elucidate the role of persistent and synaptic mechanisms in supporting working memory. By measuring the spiking of large, local populations of prefrontal neurons and resolving the dynamics of mnemonic information on single trials, we found that this information does not persist through the memory delay. Instead, cue-specific spiking activity was intermittent and was characterized by stochastically occurring, discrete transitions between robust coding of memoranda (On states) and complete lapses in such coding (Off states). Notably, these transitions were coordinated across large, local populations of neurons. Complementary to spike-rate-based coding, patterns of functional connectivity also carried information about the remembered cue during Off states. These patterns in functional connectivity are consistent with synaptic models of working memory in which cue-specific patterns of potentiated synapses facilitate the re-emergence of memory information in spike rates following silent epochs[4,5,24].

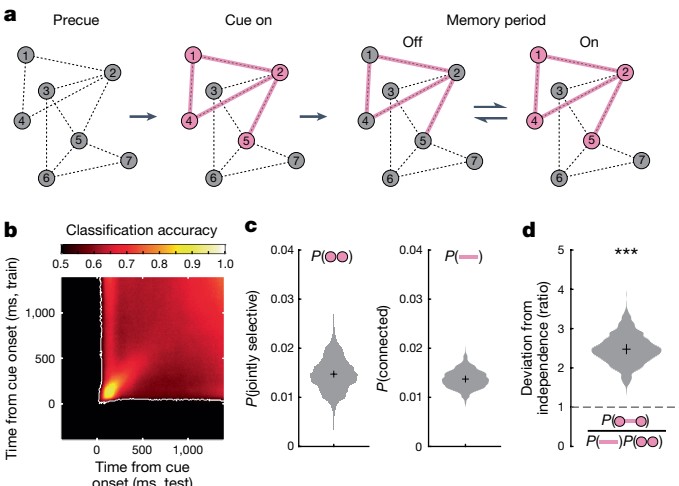

**Fig. 6 | Stability of spike coding and cue specificity of neuronal ensembles in synaptic model predictions. a**, Cartoon depicting the interplay between spiking and silent mechanisms during working memory from synaptic models of working memory. Following the before-cue period, the memory cue evokes a distinct pattern of activity among a network of neurons (circles) and a distinct pattern of STP, temporarily facilitating connectivity (lines) among cue-associated neurons. During memory delay, even in the absence of persistent-spiking activity (Off states), cue information persists in the distinct pattern of connections. During the On state, non-specific drive reignites spiking activity among cue-associated neurons. Bidirectional arrows denote stochastic transitions between Off and On states. **b**, Mean classifier accuracy ($n = 25$ sessions) plotted over time points on which the classifier was trained and tested. The block structure indicates good generalization across time. White line denotes the boundary of above-chance classification ($P < 0.001$, cluster-mass test, corrected for multiple comparisons). **c**, Left, probability ($P$) that both of a pair of neurons responded preferentially to the same cue during the evoked response (0–400 ms after cue). Right, probability that both of a pair of neurons exhibited significant CCG to a particular cue during memory delay. **d**, Proportion of pairs that were both jointly selective and connected, divided by the proportion expected. Violin plots indicate bootstrap across $n = 25$ sessions. ***$P < 0.001$, two-sided sign-rank versus one-sided ($P = 1.2 \times 10^{-5}$).

Our observations describe the dynamics of local populations of neurons in the primate lateral prefrontal cortex. Future work should explore these dynamics in other structures known to exhibit memory delay activity—for example, parietal cortex[19]—and on broader spatial scales, including the possible propagation of On states across the cortical surface[51] or distributed representations supporting memory[52,53]. Nevertheless, two facts suggest that the regions targeted by the present study play a key role in working memory. First, focal inactivation of lateral prefrontal cortex has long been known to severely impair performance in spatial working memory tasks[54–56], including selective inactivation of memory delay activity[57]. Second, in the context of our study, single-trial confidence at the onset of the behavioural response phase predicted behaviour.

Similar, large-scale electrophysiological approaches should be used to assess the relative roles of spiking and putative synaptic mechanisms across a range of working memory tasks. For example, such roles may be rather different for spatial and object-based working memory, given the apparent differences between the two in the robustness of memory delay spiking activity across brain areas[58]. Furthermore, theoretical work suggests that the relative contributions of these two mechanisms may be related to the level of manipulation of remembered information required by a task[59]. The spatial tasks used here demand a relatively straightforward sensory-to-motor transformation, and so could rely more on synaptic mechanisms.

Empirical verification of how task demands sculpt memory representations will lead to a richer understanding of the mechanistic basis of working memory.

Finally, it is interesting to note that, even within a single experimental session and task condition, the relative proportions of On and Off states showed a fair degree of heterogeneity across trials. Recent electrophysiological studies provide evidence of coordinated fluctuations in local neuronal activity in alert animals[60]. Furthermore, these coordinated fluctuations appear related to moment-to-moment changes in global arousal states, and also predict psychophysical performance in non-human primates[61,62]. Neuromodulatory inputs are among the many possible mechanisms that may contribute to coordinated fluctuations in neuronal activity[63]. Dynamics in the local tone of neuromodulators may be sufficient to induce transitions in local cortical states[60]. Within lateral prefrontal cortex, dopaminergic tone is known to play a key role in the maintenance of memory delay activity[64,65]. Thus, examining the contribution of dopaminergic tone to the variability in the dynamics of memoranda coding could prove illuminating.

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

## Methods

### Subjects

Three adult male rhesus monkeys (*Macaca mulatta*), aged 11, 12 and 8 years, participated in the experiment. Monkeys A, H and J weighed 11, 14 and 12 kg, respectively. All surgical and experimental procedures were approved by the Stanford University Institutional Animal Care and Use Committee and were in accordance with the policies and procedures of the National Institutes of Health.

### Behavioural task

Stimuli were presented on a VIEWPixx3D monitor positioned at a viewing distance of 60 cm using Psychtoolbox and MATLAB (v.R2022a, MathWorks). Eye position was monitored at 1 kHz using an Eyelink 1000 eye-tracking system (SR Research). On each trial, the animals were presented with a cue at one of eight possible locations and reported this location after a brief memory delay to receive a fluid reward. Cues were square frames (green for monkey A, black for monkey H, white for monkey J) measuring 1° of visual angle on a side, and presented at 5–7° of eccentricity (depending on the session).

Monkeys initiated behavioural trials by fixating a central fixation spot presented on a uniform grey background. After the monkeys had maintained fixation for 600–800 ms (randomly selected on each trial), a cue appeared for 50 ms at one of eight possible locations separated by 45° around fixation. Cue presentation was followed by a delay period that varied randomly from 1,400 to 1,600 ms. Following the delay period, the fixation spot disappeared and the animal was presented with one of two possible response screens. On MTS trials, two targets appeared (filled blue circles, radius 1° of visual angle (DVA)): one at the previously cued location and the other at one of the seven remaining non-cued locations. On MGS trials, no targets appeared. In either case, the animals received a reward of juice for making an eye movement to within 5 DVA of the previously cued location and then maintaining fixation for 200 ms. MTS and MGS trials were randomly interleaved such that the animals could not predict the trial type. Monkey J was trained on, and performed only, the MGS task. The animals had to maintain their gaze within either 3 DVA (monkey A) or 2 DVA (monkeys H and J) from fixation throughout the trial until the response stage. The intertrial interval was 300–1,000 ms following each correct response. Failure to acquire fixation, fixation breaks and incorrect responses were not rewarded and were followed by a 2,000 ms intertrial interval.

### Surgical procedures and recordings

Monkeys were implanted with a titanium headpost to immobilize the head, and with a titanium chamber to provide access to the brain (see ref. 12 for full details). In a previous study[66], we identified the frontal eye field based on its neurophysiological characteristics and ability to evoke saccades with electrical stimulation at low currents[67]. Here we recorded from both area 8, within and anterior to the frontal eye field, and the principal sulcus (9/46) (monkey A, area 8; monkey H, areas 8 and 9/46; monkey J, area 9/46), using primate Neuropixels probes[39]. During each session, we pierced the dura using a screw-driven, 21-gauge pointed cannula and lowered a single probe through this cannula using a combination of custom three-dimensional printed grids and motorized drives (NAN instruments). Probe trajectories spanned several cortical columns, as inferred from the broad distribution of preferred cue locations across neurons. Recordings were allowed to settle for around 30 min before the start of the experiment, to mitigate drift. We configured probes to record from 384 active channels in a contiguous block, allowing dense sampling of neuronal activity along a 3.84 mm span.

Neuropixels filter and digitize activity at the headstage separately for the action potential bands (300 Hz high-pass filter, 30 kHz sampling frequency) and local field potential (1 kHz low-pass filter, 2.5 kHz sampling frequency). Activity was monitored during experimental sessions and saved to disk using SpikeGLX (https://billkarsh.github.io/SpikeGLX/).

### Data preprocessing

Spiking in the action potential band was identified and sorted offline using Kilosort3 (ref. 68). Because we were interested in population-level coding of memory, we analysed both putative single- and multi-unit clusters identified by Kilosort. Spike times were aligned to a digital trigger on each trial, indicating cue onset, and corrected for a lag in stimulus presentation estimated offline using photodiode measurements from the stimulus display and the timing of the cue-evoked response. Neurons that fired fewer than 1,000 spikes in the experimental sessions (each roughly 3 h) were excluded from further analyses. Spike times were converted to smoothed firing rates (sampling interval, 10 ms) by representing each spiking event as a delta function and convolving this time series with a 100 ms boxcar. For CCG analyses, unsmoothed spikes times were binned with a width and timestep of 1 ms. Incorrect trials were rare (Fig. 1b) and were excluded from subsequent analysis. Waveform templates were localized in space using NeuropixelUtilities (https://djoshea.github.io/neuropixel-utils/). Local field potentials (LFPs) were sampled at 2,500 Hz. Offline, LFPs were filtered using a 2–200-Hz-bandpass, zero-phase Butterworth, notch filtered at 60 Hz and downsampled to 1,000 Hz. LFPs were transformed into the time-frequency domain using Morlet wavelets and downsampled to 100 Hz.

### Statistics and reproducibility

All statistical tests are two-sided unless otherwise specified. Key findings and brain–behaviour relationships were evident in each of the three animals (Extended Data Figs. 2 and 7), and in each of the 25 individual experimental sessions (Extended Data Fig. 3). Sample size (three animals, 8,255 neurons) was chosen based on standards in the field. Each animal was exposed to every task manipulation. Within a session, task manipulations were randomized across trials. Neurons were recorded without bias, and electrodes placed to maximize signal-to-noise of the electrophysiological signal.

### Functional subtyping

To determine the functional subtype of units[69] (Extended Data Fig. 1), we analysed firing rates during three time epochs: visual (0–300 ms after cue onset), memory (500–1,400 ms after cue onset) and motor (100–300 ms post-fixation offset). A unit was labelled as being selective during a given epoch if firing rates during that epoch were significantly modulated by cue location (one-way ANOVA, $P < 0.05$ criterion). Units were then sorted into functional subclasses based on the set epochs during which each unit was selective.

### Classification of cue location

Firing-rate estimates for each unit and time point relative to cue onset were $z$-scored across trials before classification. We used linear classifiers to quantify the amount of information on the location of the cue in populations of simultaneously recorded units. We held out each trial for test one by one, training a logistic regression classifier (as implemented by fitclinear.m in MATLAB) to predict cue location using the population vector of firing rates. Specifically, classifiers were trained to discriminate the same cue location as the test trial from the opposite cue location, using the applicable subset of trials from the training set. Data were subsampled during training to equalize trial counts for the two conditions. A unique classifier was trained and tested for each time point relative to cue onset. 'Classification accuracy' reflects the proportion of correctly classified test trials (Fig. 2c); 'classifier confidence' is the non-thresholded value of the logistic function corresponding to the probability assigned by the classifier to the correct label at test (Fig. 3).

Cross-state classification (Extended Data Fig. 6a) was similar, except that only values from trial time points labelled On or Off (as appropriate) entered the training set or were held out as a test trial. Test confidence

values were averaged across the memory delay (500–1,400 ms after cue) to yield the final results.

Cross-temporal classification (Fig. 6b) was also similar, except that we used a split-half approach in which the classifiers for each time point were trained on half of the available population of trials and tested (cross-temporally) using the other half.

## Mixture modelling of confidence

We used a mixture-modelling approach to test whether confidence during the memory delay (500–1,400 ms after cue) was best described as drawn from a one- or a two-state distribution (Extended Data Fig. 5). To do this, for each session and cue location we modelled the probability density function of confidence values during the memory delay as either a single beta distribution,

$$p(c) = \text{Beta}(c; \alpha, \beta),$$

or a mixture of two beta distributions,

$$p(c) = w \times \text{Beta}(c; \alpha, \beta) + (1 - w) \times \text{Beta}(c; \alpha', \beta'),$$

where $c$ is confidence, $\alpha$, $\beta$, $\alpha'$ and $\beta'$ parameterize beta distribution(s) and $w$ is the mixing coefficient. The best-fitting parameters of each model were identified by maximum-likelihood estimation using gradient descent in MATLAB. We used fourfold cross-validation on the population of trials to assess the likelihood of each model on held-out test data, and then normalized by the number of trials and changed the log likelihood to base 2 to yield the cross-validated score of each model in terms of bits per trial. Finally, we subtracted these two model scores and averaged across conditions to yield the difference in model performance for each session.

Note that our choice of beta distribution here is principled: it is extremely flexible, able to demonstrate a broad range of skewness and kurtosis and naturally accommodates bounded continuous variables such as confidence[70]. This flexibility makes this analysis conservative, ensuring that the one-state model is capable of describing a broad range of empirical distributions. Nevertheless, results were not dependent on the exact modelling approach: similar results for our one-versus-two-state model comparison were obtained when using either Gaussian mixture or hidden Markov models.

## Analysis of microsaccades

The horizontal and vertical eye position records were convolved with a Gaussian kernel ($\sigma = 4.75$ ms) to suppress noise before taking first derivatives, yielding the eye velocity along each dimension. We then took the root sum of squares of the horizontal and vertical velocities to obtain eye speed. We flagged peaks in this time series with a minimum peak height of $10°\ s^{-1}$ and a minimum interpeak distance of 50 ms as microsaccades (ref. 71 and Extended Data Fig. 4), which were confirmed by visual inspection of the data.

## Labelling of On and Off states

To identify On and Off states (Fig. 4a), we repeated the cue classification analysis described above 50 times, randomly shuffling the labels of the training set for each test trial. This yielded, for each trial, a null distribution of 50 confidence time series (Fig. 4a). We then z-scored each time point of the true confidence time series by the mean and standard deviation of this null distribution. Individually significant (above 1.96) z-values were cluster corrected for multiple comparisons over time[72]. In brief, we compared the sum of contiguous individually significant z-values with that expected by chance (randomization test). Clusters with a mass greater than the 95% percentile of the null were labelled On states; contiguous z-values falling below a conservative ($P > 0.20$) threshold for at least five consecutive time points were labelled Off states.

## Tuning curves

To test whether On and Off states reflected coordinated changes in tuning across the neural population, we used a split-half approach. First, firing-rate estimates for each unit and time point relative to cue onset were z-scored across trials. Then, for each session, we randomly divided the population of units in half. We used one half of the units to identify On and Off states, as described above. Next, for each unit in the held-out population, we computed mean firing rate during the memory delay for each cue location separately for On and Off states, averaging across relevant time points and across trials. This yielded, for each unit, two eight-element vectors—the On and Off tuning functions. To align tuning functions across units, the preferred cue location for each was identified as the condition in which the sum of the On and Off functions was greatest, and assigned an arbitrary value of zero degrees. Alignment of tuning curves to the maximum-valued preferred cue in this way will necessarily produce a peak at zero degrees in the average tuning function, even in the absence of true tuning. To correct for this, for each unit we also computed null On and Off tuning functions by first shuffling cue labels across trials, aligned these to the preferred cue and subtracted them from the true On and Off tuning functions (Fig. 3d).

For demonstration purposes, we fit the average On and Off tuning functions with a difference of Gaussians using gradient descent in MATLAB. Difference of Gaussians is useful for describing tuning curves that show surround suppression[73].

## Population firing rates

To describe how population firing rates evolved over the course of the trial, we averaged these across all units recorded in the same session and across all trials for the preferred cue location (greatest mean classification confidence during the memory delay), yielding a single time series for each session. We then normalized this time series by the mean and standard deviation of a 400 ms baseline period (−400 to 0 ms relative to cue onset), yielding a metric of population spiking in units of standard deviations above baseline (Fig. 4e, grey traces). We repeated this analysis for the memory delay, this time including only data points labelled On or Off (Fig. 4e, orange and blue traces).

## Phase–state relationships

To determine whether the phases of different frequency components of the LFP were predictive of On and Off states, we first extracted the phase from the time-frequency representation of the LFP. Next, we identified the onset time of On and Off states during the memory delay across all trials within a session. Then, for each session, probe channel and frequency (4–60 Hz), we computed the (circular) mean phase at On state and Off state onset, and the magnitude of the angular difference between these means. If phase is predictive of state, this difference should be larger than that expected by chance. Accordingly, we obtained a null distribution of phase difference magnitudes by repeating this procedure 1,000 times, randomly permuting On and Off labels for each phase measurement on each iteration and used this null distribution to generate a z-score metric. Z-scores were averaged across channels, yielding a phase–state metric for each session and frequency of interest. Finally, we tested whether these scores were greater than zero for each frequency of interest (cluster-corrected randomization test).

## Standard deviation of background noise

To ensure that changes in recording quality could not account for the presence of On and Off states in our recordings, we measured the standard deviation of background noise[42] in the action potential band (0.3–10 kHz). Specifically, for each On and Off state identified using the

non-parametric procedure described above, noise standard deviation was estimated as

$$\sigma_n = \mathrm{median}\left(\frac{|x|}{0.6745}\right),$$

where $x$ is the time series of raw action potential band values recorded during the state. Noise estimates were averaged across all On and Off states within each session.

### Demixed principal components analysis

We used demixed principal components analysis analysis to decompose population activity into different components reflecting cue location, time and their interaction. As with our classification-based analyses, we applied demixed principal components to the smoothed firing rates from each session, focusing on activity during the delay period (500–1,400 ms after cue). The proportion of variance explained and components were extracted as described in ref. 50.

### Single-neuron ANOVA

We downsampled the smoothed firing rates for each neuron to 100 ms steps and modelled firing rate during the delay (500–1,400 ms after cue) as a linear combination of cue location, firing rate and their interaction, to estimate the proportion of variance explained by each of these terms.

### CCG analysis

To characterize functional connectivity among units, we computed cross-correlations between spike trains of all pairs of simultaneously recorded neurons with mean firing rates greater than 1 Hz. CCGs were computed separately for each cue location. Following previous studies[46], to mitigate firing-rate effects, we normalized cross-correlation for each pair of neurons by the geometric mean of their firing rates for the cue location condition under consideration. The CCG for a pair of neurons ($j, k$) in condition $c$ was therefore

$$\mathrm{CCG}(\tau)_{j,k,c} = \frac{\sum_{i=1}^{M}\sum_{t=\tau+1}^{N} x_j^i(t-\tau) \times x_k^i(t)}{\sqrt{\sum_{i=1}^{M}\sum_{t=\tau+1}^{N} x_j^i(t-\tau) \times \sum_{i=1}^{M}\sum_{t=\tau+1}^{N} x_k^i(t)}},$$

where $M$ is the number of trials collected for cue location $c$, $N$ is the number of time bins within a trial, $\tau$ is the time lag between the two spike trains and $x_k^i(t)$ is 1 if neuron $j$ is fired in time bin $t$ of trial $i$, but zero otherwise.

To correct for correlation due to stimulus locking or slow fluctuations in population response, we subtracted a jittered CCG from the original. This jittered CCG reflects the expected value of the CCG computed from all possible jitters of each spike train within a given jitter window[74,75]. The jittered spike train preserves both the poststimulus time histogram (PSTH) of the original spike train across trials and the spike count in the jitter window within each trial. As a result, jitter correction removes the correlation between PSTHs (stimulus locking) and those on time scales longer than the jitter window (slow population correlations). We chose a 25 ms jitter window, following previous work[45,46,76,77].

We classified a CCG as significant if the peak of the jitter-corrected CCG occurred within 10 ms of zero and was more than seven standard deviations above the mean of a high-lag baseline period (100 > $|\tau| > 50$)[45]. Zero-lag CCGs were excluded from the analyses reported here, although their inclusion yielded statistically indistinguishable results.

All CCGs were estimated using spike trains during the memory delay (500–1,400 ms after cue) to avoid the influence of visually evoked responses. CCG analyses specific to On and Off states (Fig. 5f) were computed by first setting $x(t)$ to zero for all time points not identified as On or Off (respectively), and then repeating the analysis described above.

### Manhattan distance

To determine whether patterns of functional connectivity differed according to the contents of memory, we compared the graphs of significant CCGs across cue locations in a pairwise manner (Fig. 5c–f). For each session and cue location, we represented the results of our CCG analyses as a graph in which nodes were units. The edge (connection) between each pair of units was assigned a weight of 1 if the pair had a significant CCG, and zero otherwise. Then, for each possible pair of cue locations, we computed the Manhattan distance, the number of edges with a weight that differed across the two graphs. Finally, we averaged this metric across all 28 possible pairs of conditions, yielding one summary statistic per session.

To normalize this mean Manhattan distance for comparison across sessions, we shuffled the cue location labels within each pair of neurons for each pair of conditions under consideration across trials, and repeated the entire analysis pipeline 50 times (25 for analyses specific to On and Off states), from CCG estimation through Manhattan distance calculation. We then $z$-scored the mean Manhattan distance for each session by this null distribution and compared these $z$-scores to zero (Fig. 5f).

Note that CCGs among both single and multi-units have been widely used as a measure of functional connectivity[78–84]. Indeed, CCGs based on multi-unit activity may be more sensitive in detection of correlations in spiking than similar analyses of single-neuron pairs[78,85,86]. Nonetheless, the presence of multi-units in our dataset does limit the conclusions that might be drawn about the specific neuronal subtypes involved in the cue-dependent ensembles that we observe—for example, putative pyramidal versus non-pyramidal neurons.

### Firing-rate-matched control

The geometric mean firing rate of pairs of units varied significantly across the eight cue locations (one-way ANOVA, $P = 0.002$; Fig. 6a). Geometric mean firing rates were statistically indistinguishable, however, across cue locations 1–4 ($P = 0.332$) and 5–8 ($P = 0.884$). Therefore, we repeated the analysis of Manhattan distance described above, this time computing it among only cue locations 1–4 and 5–8 (Extended Data Fig. 6b), to yield a firing-rate-matched variant of the analysis presented in Fig. 5f.

### Joint selectivity

To determine the selectivity of units during the evoked response, we averaged each unit's cue-locked firing rate over time (0–400 ms after cue onset), yielding an nTrials × 1 vector of firing rates. We then performed one-way ANOVA to evaluate the relationship between cue location and firing rate. If the effect of cue location was significant ($P < 0.05$), the unit was deemed selective to cue location, and the location at which it had the greatest mean firing rate was labelled the preferred location. Pairs of units were deemed jointly selective if they were selective for the same cue location.

### Reporting summary

Further information on research design is available in the Nature Portfolio Reporting Summary linked to this article.

## Data availability

The data underlying this study are available at Dryad (https://doi.org/10.5061/dryad.kkwh70sct)[87].

## Code availability

The code underlying this study is available at GitHub (https://github.com/panichem/SingleTrialDynamics).

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

**Acknowledgements** We thank D. Lopes and S. Cital for veterinary assistance, M. Solyali and B. Schneeveis for machining assistance and T. Engel, S. Druckmann, A. Soltani and X. Chen for helpful comments. This work was supported by NIH grant nos. EYO14924 and NS116623, and by a Ben Barres Professorship to T.M.

**Author contributions** M.F.P. and T.M. conceived the project. D.J. and T.M. designed the tasks. D.J. and M.P. trained the animals. T.M., D.J. and M.F.P. carried out surgical procedures. M.F.P. and D.J. collected data. M.F.P. and J.Y.O. analysed the data. S.Z. and E.B.T. provided code for computing CCGs. T.M. supervised the work. T.M. acquired funding. M.F.P. and T.M. wrote the first draft. All authors reviewed and edited the manuscript.

**Competing interests** The authors declare no competing interests.

**Additional information**
**Correspondence and requests for materials** should be addressed to Matthew F. Panichello or Tirin Moore.

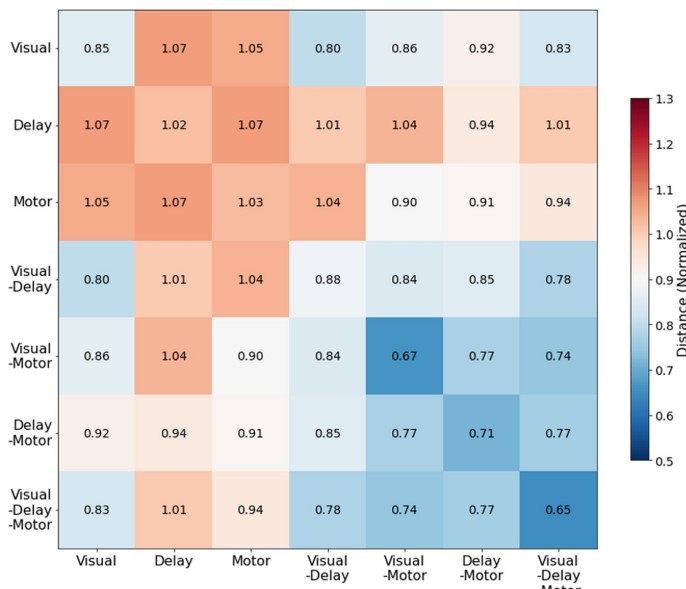

**Extended Data Fig. 1 | Normalized spatial distance between different functional classes of neurons within lateral prefrontal cortex.** Seven functional classes of neurons were defined according to selective activity to three task components: visual, delay, and motor. Neurons were defined as having a given functional property based on the presence of significant selectivity across cue conditions within the visual, delay and motor epochs of the task. The plot shows the mean distance between different classes of neurons; means were calculated and normalized by each session's total mean distance. Top-left to bottom-right diagonal elements show the mean distance within each functional class. Interestingly, we did not observe a relationship between depth and different functional subclasses (not shown), although this result should be interpreted with caution due to our non-perpendicular electrode penetrations.

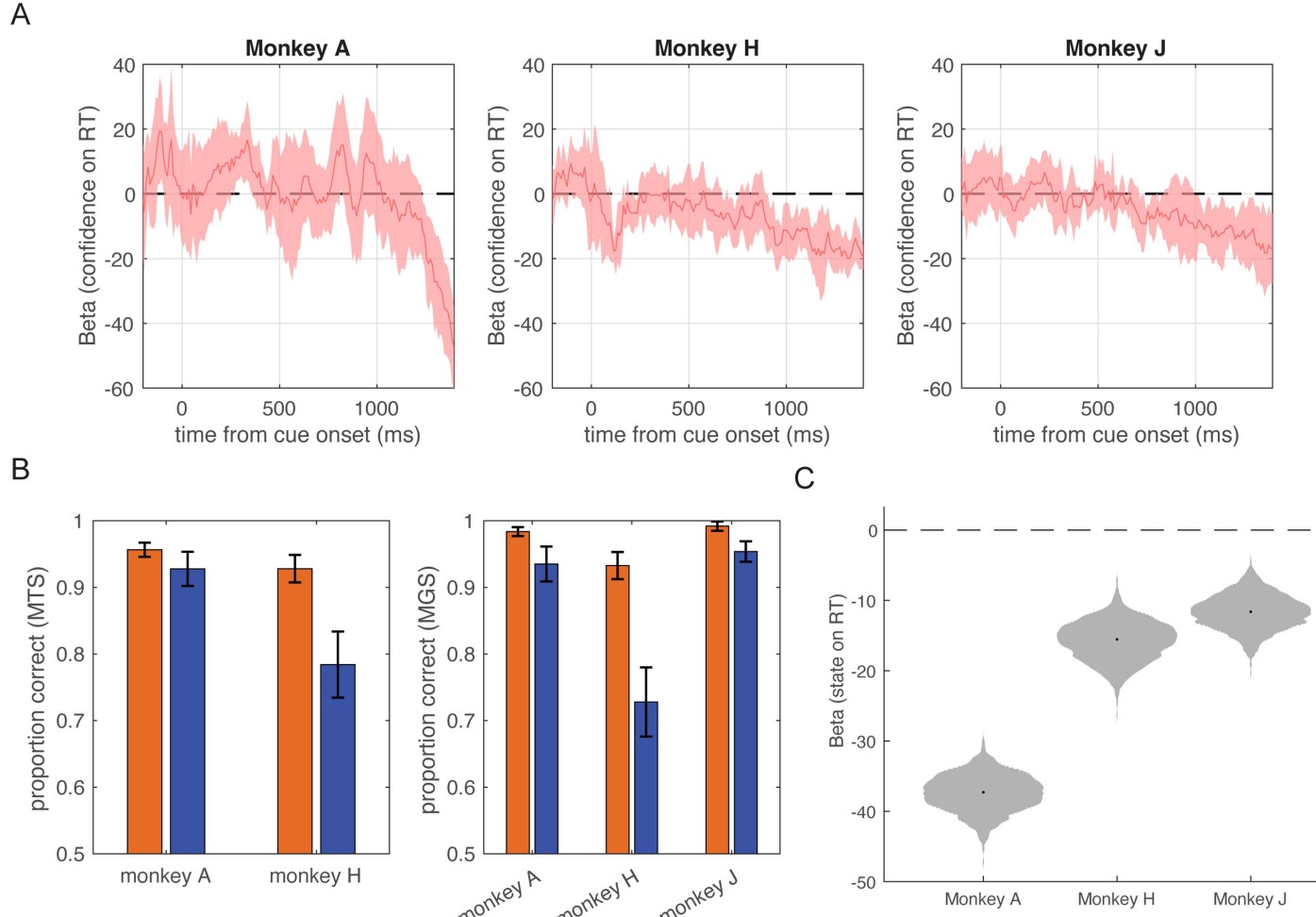

**Extended Data Fig. 2 | Confidence and state predict behavior.** (A) Regression coefficient relating classifier confidence to reaction time in milliseconds (average across sessions). Cue location and task (MTS/MGS) were included as co-regressors. Shaded area: 95% confidence intervals. (B) Mean proportion correct in the delayed match-to-sample (MTS) and memory-guided saccade (MGS) tasks, binned by state (orange = On, blue = Off) immediately after fixation offset. Error bars are 95% confidence intervals (binomial). (C) Regression coefficient relating state (On/Off) immediately after fixation offset to reaction time in milliseconds. Cue location and task (MTS/MGS) were included as co-regressors. Negative values indicate that On states are associated with reduced reaction times. Violin plots show bootstrap across sessions.

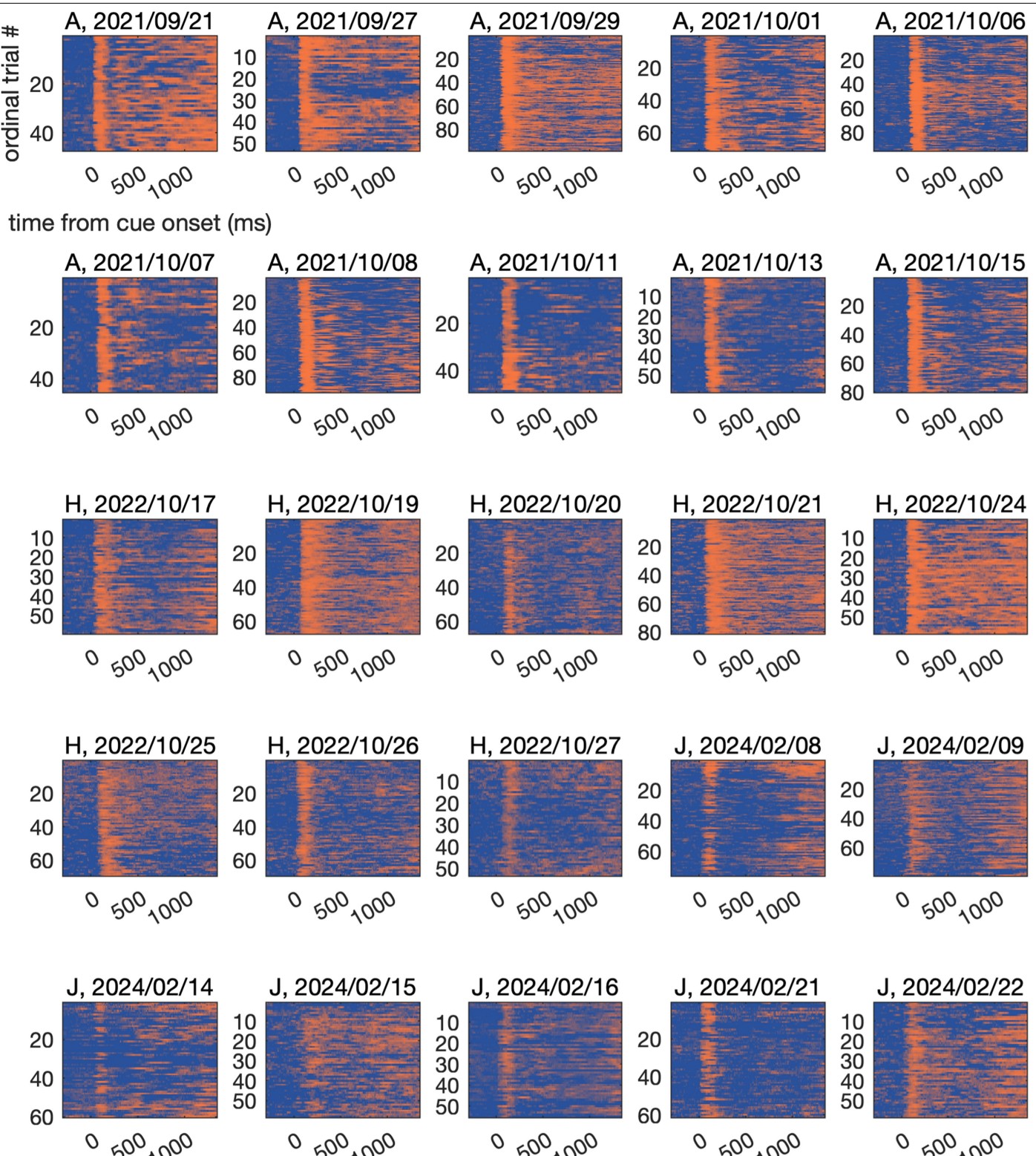

**Extended Data Fig. 3 | Single-trial classifier confidence, relative to cue onset, for all trials from the most preferred cue condition for each session.** Color scale as in Fig. 2a.

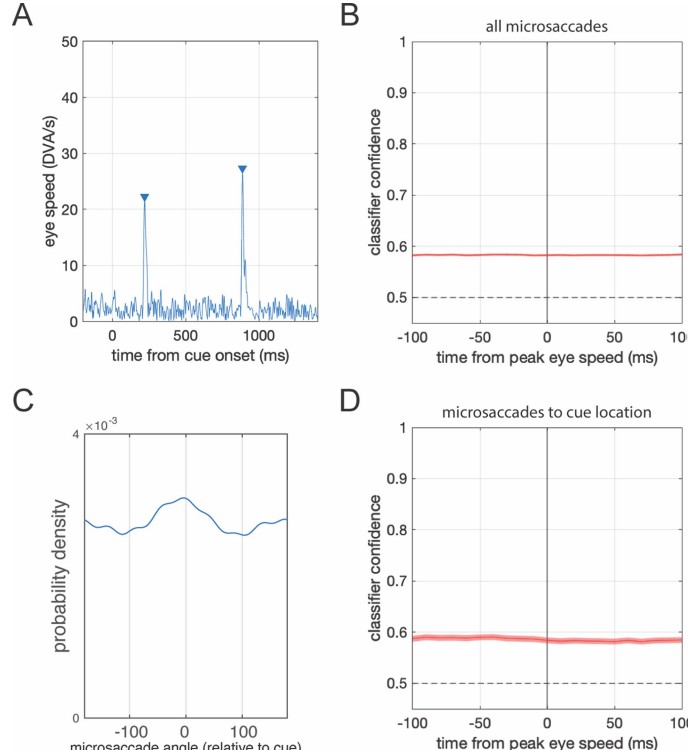

**Extended Data Fig. 4 | Classifier confidence was not affected by microsaccades.** (A) Example trial showing eye speed data and microsaccade identification. Microsaccades were identified as peaks in eye speed >10 DVA/s. (B) Mean classifier confidence during the memory delay, locked to microsaccades (average across N = 8,910 microsaccades). Shaded area (small) indicates 95% confidence intervals. (C) Probability density of microsaccade angles during the memory delay, relative to the cued location. (D) Mean classifier confidence during the memory delay, locked to microsaccades to the cued location (+/− 20 degrees).

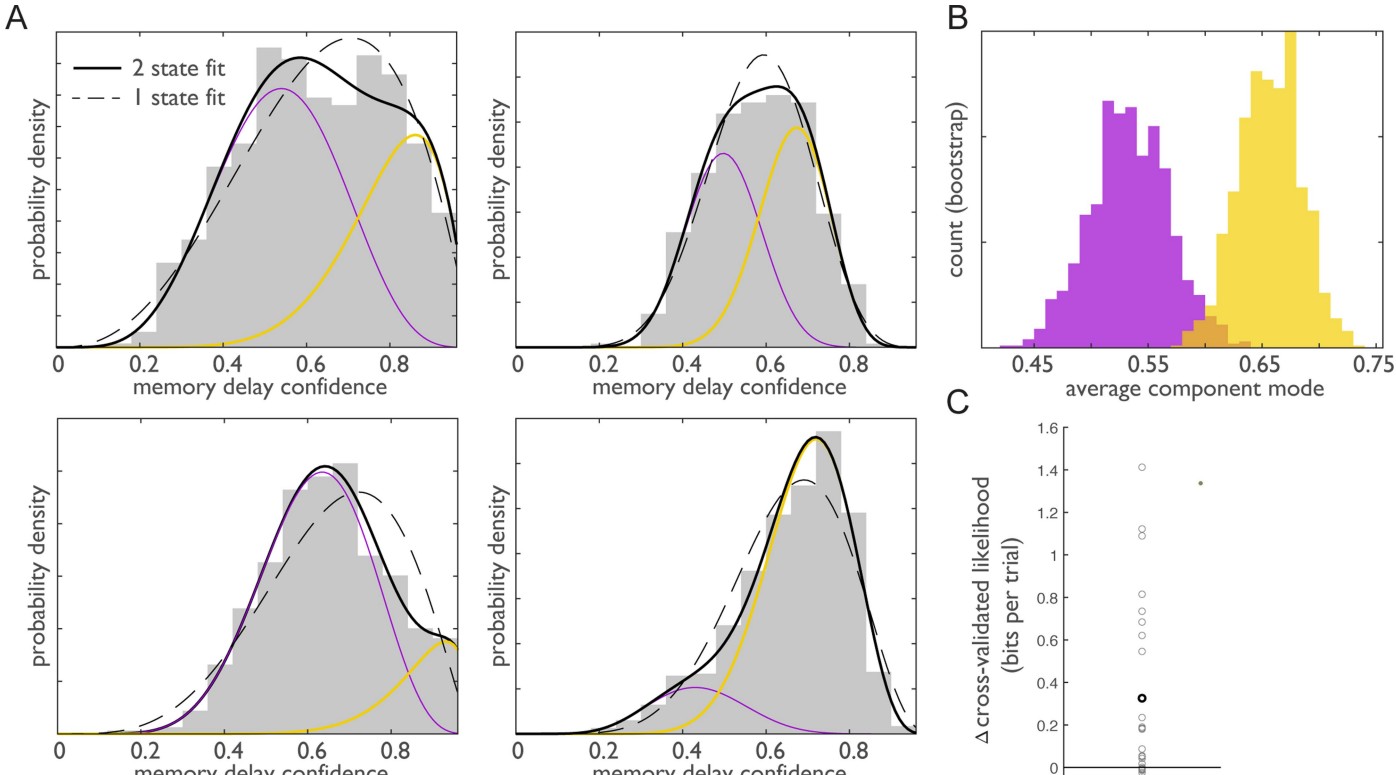

**Extended Data Fig. 5 | Mixture modeling of behavior.** (A) Histogram of confidence values during memory delay time points (one cue location) from three example sessions. Dashed line shows the best fitting beta distribution. Solid line shows the best fitting mixture of two beta distributions. Purple and yellow lines show the individual component distributions for the two-state model. (B) Histogram showing average proportion-weighted component modes (bootstrap). (C) Cross-validated model comparison results for 2-state vs 1-state fits for all sessions (N = 25). The 2-state model outperforms the 1-state (p = 0.009, chi-squared). Light circles show individual session scores; dark circle shows mean across sessions; black line shows equivalent model performance.

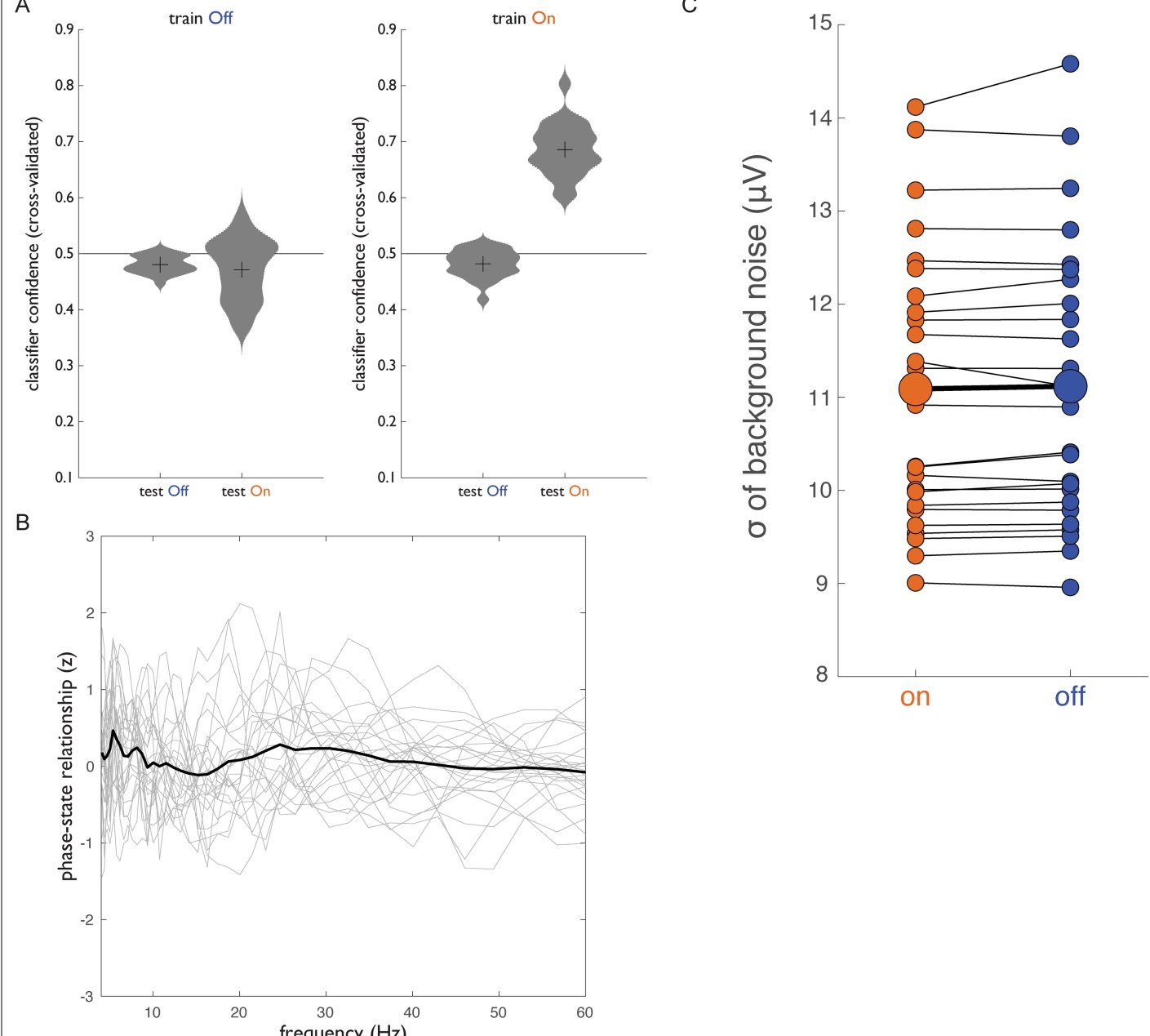

**Extended Data Fig. 6 | Off states were not explained by an alternate population code, rhythmic activity, or background noise.** (A) Cross-validated classifier confidence, binned by training and test state. Left: classifier confidence when trained on Off states and tested on Off and On states. Right: classifier confidence when trained on On states and tested on Off states. Violin plots show bootstrap across sessions. (B) Mean phase-state relationship (z-scored) as a function of frequency. Gray traces show individual sessions (N = 25). Values reflect absolute angular difference between the mean phase of On and Off states, normalized by the null (randomization test). (C) Standard deviation of the background noise, in microvolts, averaged across all On and Off states. Small circles: individual session means; large circles: grand means. Background noise did not differ across On and Off states (p = 0.339, sign-rank).

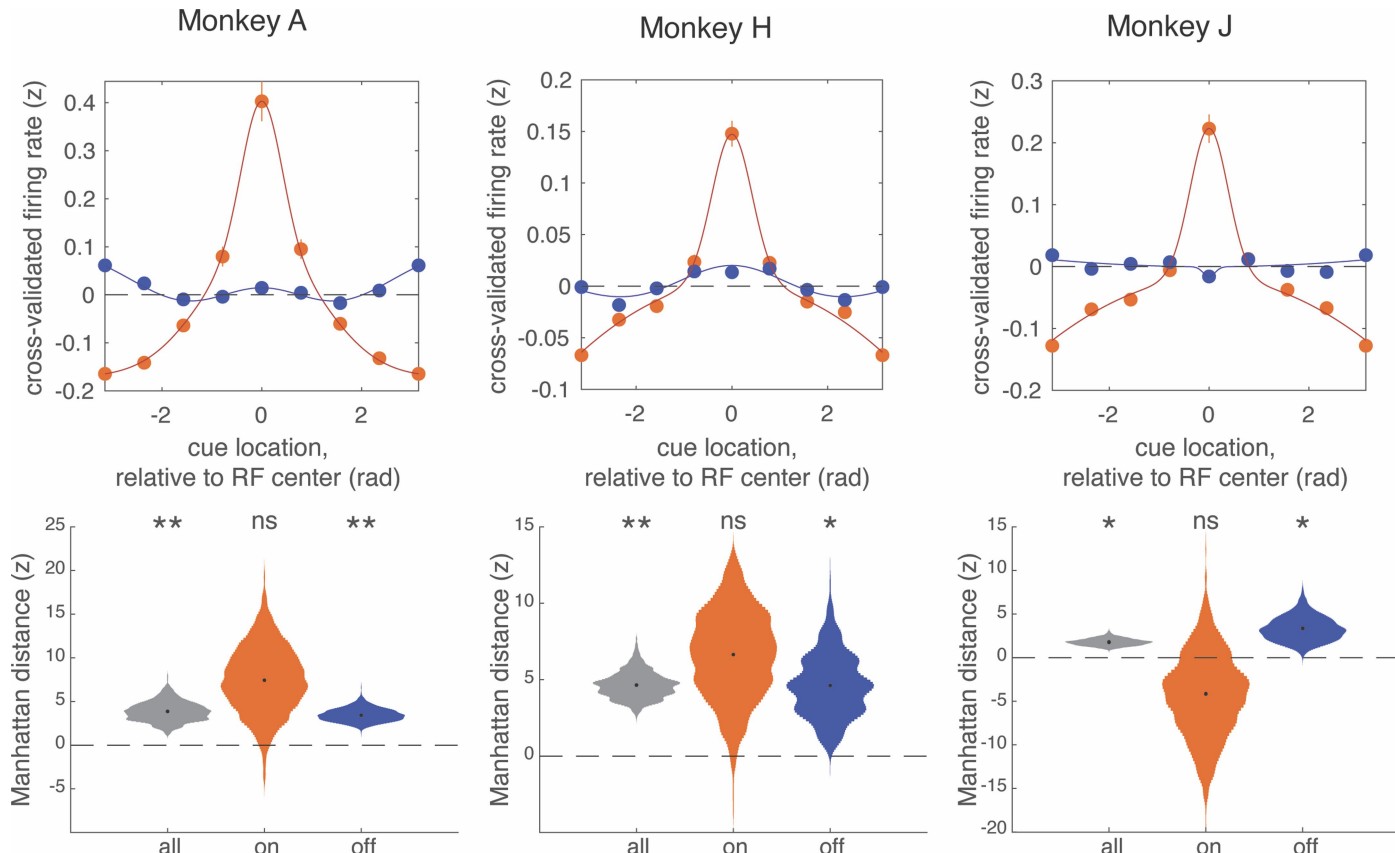

**Extended Data Fig. 7 | Results for each animal.** Top: Single-animal memory tuning functions for held-out units during On and Off states. Tuning functions show the mean normalized firing rate during the memory delay (z-scored across trials) for held-out units, relative to each unit's preferred cue location. Error bars (small) denote SEM. Bottom: Single-animal mean normalized Manhattan distance, using data from the entire memory delay (gray), only during On states (orange), and only during Off states. Violin plots show bootstrap across sessions. Asterisks and 'ns' denote significance.

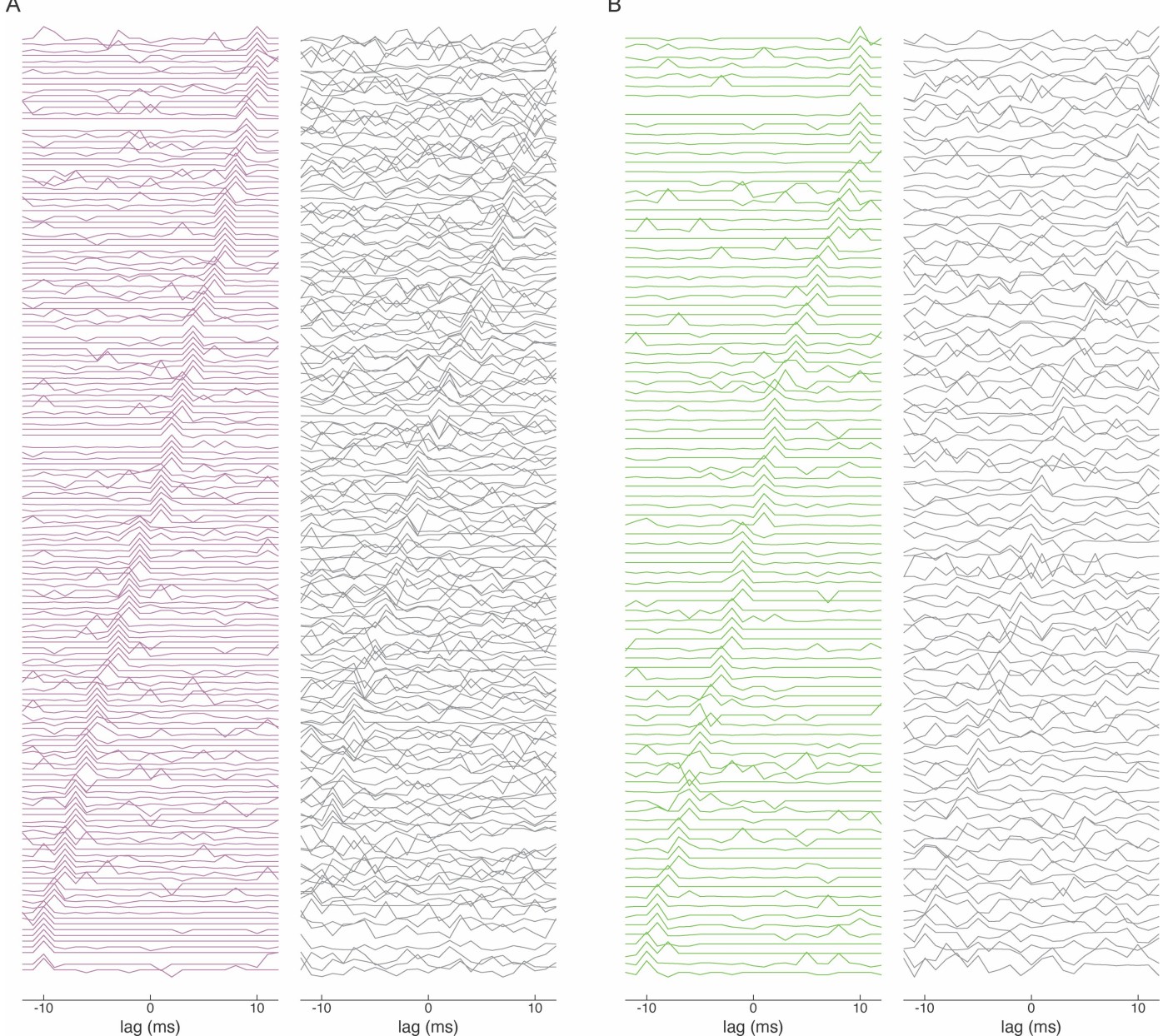

**Extended Data Fig. 8 | Example cross-correlograms (CCGs) with significant and non-significant peaks.** (A) All CCG functions from the "left" cue condition with a significant non-zero lag peak (pink), along with an equal number of randomly selected "non-significant" CCGs (grey) from that same cue condition. (B) Same as in A, for the "right" cue condition. All CCGs in each subplot are arranged by the lag of their maximum value.

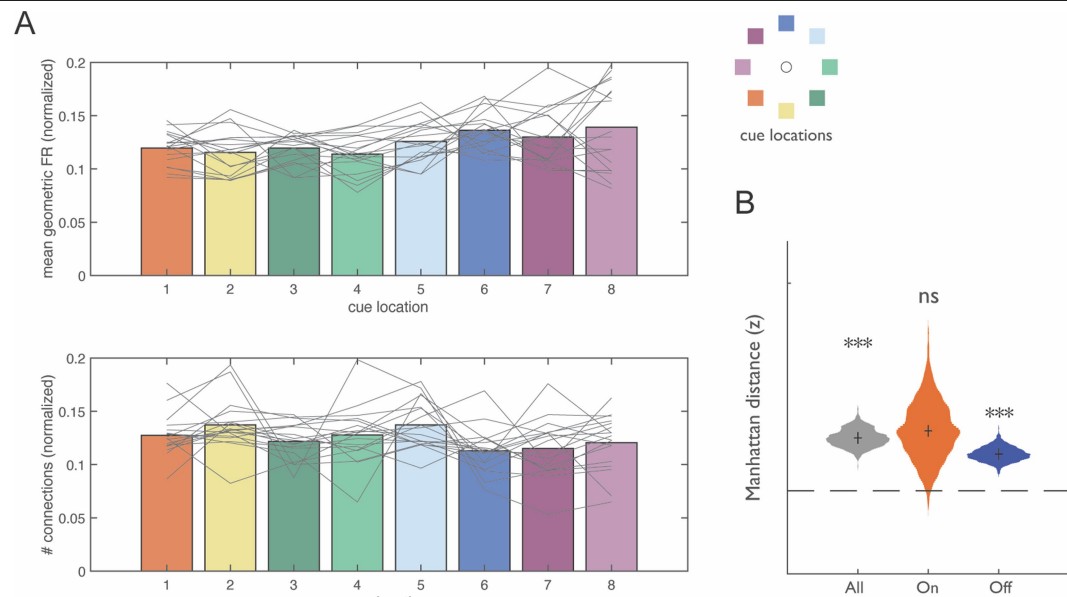

**Extended Data Fig. 9 | Firing rate matching for CCG analysis.** (A) Top: Normalized geometric mean firing rate (gFR) of all pairs of neurons during the memory delay. Bottom: Normalized number of neuronal pairs with significant CCGs. Gray lines show individual sessions; bars show mean across sessions; colors indicate cue location (lower inset). (B) Mean normalized Manhattan distances restricted to a comparison among cue conditions (1-4 and 5-8) in which gFR was equal (see Methods). Shown are means of data from all sessions during the entire memory delay (gray), only during On states (orange), and only during Off states (blue). Violin plots show bootstrap across sessions.

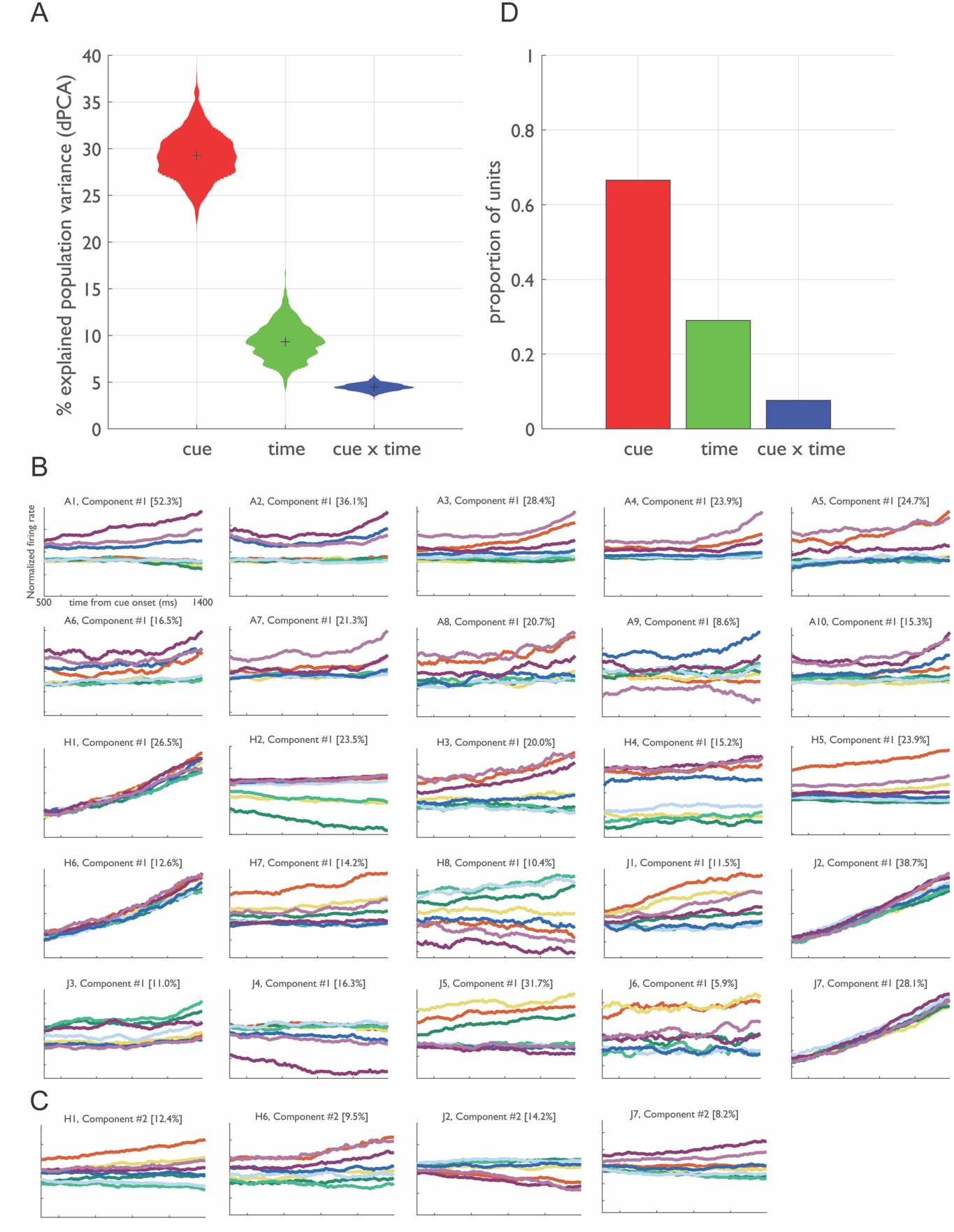

**Extended Data Fig. 10** | See next page for caption.

**Extended Data Fig. 10 | Trial-averaged rate coding is predominantly stable over the memory delay.** (A) Proportion of variance explained by each of the task components (dPCA). Violin plots reflect bootstrap across sessions. (B) Stable coding was also evident in the demixed components identified by dPCA. As with conventional PCA, each of these components can be understood as a linear combination (or, critically, a linear readout) of the population of trial-averaged single-neuron PSTHs. For 21 of the 25 total experimental sessions, the dPCA component explaining the most variance in the data encoded the cue in a time-invariant manner. The largest component from the remaining 4 sessions was a cue-invariant ramp unrelated to mnemonic coding; for these sessions (C), the second component captured time-invariant information about the cue. Consistent with the analysis of explained variance, none of these components displayed sequential coding or other time-varying signals. (D) Proportion of neurons displaying a significant effect of cue, time, and their interaction during the memory delay.

Tirin Moore

# Reporting Summary

## Statistics

For all statistical analyses, confirm that the following items are present in the figure legend, table legend, main text, or Methods section.

| n/a | Confirmed | |
|---|---|---|
| ☐ | ☒ | The exact sample size (*n*) for each experimental group/condition, given as a discrete number and unit of measurement |
| ☐ | ☒ | A statement on whether measurements were taken from distinct samples or whether the same sample was measured repeatedly |
| ☐ | ☒ | The statistical test(s) used AND whether they are one- or two-sided<br>*Only common tests should be described solely by name; describe more complex techniques in the Methods section.* |
| ☒ | ☐ | A description of all covariates tested |
| ☐ | ☒ | A description of any assumptions or corrections, such as tests of normality and adjustment for multiple comparisons |
| ☐ | ☒ | A full description of the statistical parameters including central tendency (e.g. means) or other basic estimates (e.g. regression coefficient) AND variation (e.g. standard deviation) or associated estimates of uncertainty (e.g. confidence intervals) |
| ☐ | ☒ | For null hypothesis testing, the test statistic (e.g. *F*, *t*, *r*) with confidence intervals, effect sizes, degrees of freedom and *P* value noted<br>*Give P values as exact values whenever suitable.* |
| ☒ | ☐ | For Bayesian analysis, information on the choice of priors and Markov chain Monte Carlo settings |
| ☒ | ☐ | For hierarchical and complex designs, identification of the appropriate level for tests and full reporting of outcomes |
| ☒ | ☐ | Estimates of effect sizes (e.g. Cohen's *d*, Pearson's *r*), indicating how they were calculated |

*Our web collection on statistics for biologists contains articles on many of the points above.*

## Software and code

Policy information about availability of computer code

| | |
|---|---|
| Data collection | Stimuli were presented and behavioral responses registered using Psychtoolbox (version 3) and Matlab R2022a. Code for stimulus presentation is available at https://github.com/panichem/SingleTrialDynamics. Neuropixels data were acquired using SpikeGLX. Waveforms were sorted using Kilosort3. |
| Data analysis | Data were analyzed using built-in functions and custom code written in Matlab. Built-in functions are noted in the associated methods section (e.g., fitclinear for logistic regression). Equations for non-standard statistics are provided in the methods.<br>Code for custom functions are provided in the manuscript as a reference to the original source or are available on GitHub (https://github.com/panichem/SingleTrialDynamics). |

For manuscripts utilizing custom algorithms or software that are central to the research but not yet described in published literature, software must be made available to editors and reviewers. We strongly encourage code deposition in a community repository (e.g. GitHub). See the Nature Portfolio guidelines for submitting code & software for further information.

## Data

Policy information about availability of data

All manuscripts must include a data availability statement. This statement should provide the following information, where applicable:
- Accession codes, unique identifiers, or web links for publicly available datasets
- A description of any restrictions on data availability
- For clinical datasets or third party data, please ensure that the statement adheres to our policy

The data underlying this study are available on Dryad (https://doi.org/10.5061/dryad.kkwh70sct).

## Research involving human participants, their data, or biological material

Policy information about studies with human participants or human data. See also policy information about sex, gender (identity/presentation), and sexual orientation and race, ethnicity and racism.

| | |
|---|---|
| Reporting on sex and gender | n/a |
| Reporting on race, ethnicity, or other socially relevant groupings | n/a |
| Population characteristics | n/a |
| Recruitment | n/a |
| Ethics oversight | n/a |

Note that full information on the approval of the study protocol must also be provided in the manuscript.

# Field-specific reporting

Please select the one below that is the best fit for your research. If you are not sure, read the appropriate sections before making your selection.

☒ Life sciences ☐ Behavioural & social sciences ☐ Ecological, evolutionary & environmental sciences

For a reference copy of the document with all sections, see nature.com/documents/nr-reporting-summary-flat.pdf

# Life sciences study design

All studies must disclose on these points even when the disclosure is negative.

| | |
|---|---|
| Sample size | As detailed in the manuscript, a total of 8,225 single and multi-unit neurons were recorded from 3 animal subjects across 25 experimental sessions. The number of subjects (3) is above the standard in the field (Fries & Maris, 2022). The number of neurons is in line with the current state-of-the-art (Trautmann et al., 2023). |
| Data exclusions | Neurons with very low firing rates were excluded from further analysis. Exclusion criteria are detailed in the methods. |
| Replication | Independent experiments were performed in 3 animals. All data were analyzed, except for the exclusion criteria described above. There were no failed replication attempts (i.e., no animals failed to learn the task and no neural recordings were excluded). |
| Randomization | Each animal was exposed to every task manipulation. Within a session, task manipulations were randomized across trials. Neurons were recorded without bias, with electrodes placed to maximize signal-to-noise of the electrophysiological signal. |
| Blinding | All animals were assigned to a single experimental group, and so blinding was not necessary or possible. However, experimenters were blinded to experimental conditions during recording of neurons and during sorting of waveforms into single neurons. |

# Reporting for specific materials, systems and methods

We require information from authors about some types of materials, experimental systems and methods used in many studies. Here, indicate whether each material, system or method listed is relevant to your study. If you are not sure if a list item applies to your research, read the appropriate section before selecting a response.

## Materials & experimental systems

| n/a | Involved in the study |
|-----|----------------------|
| ☒ | ☐ Antibodies |
| ☒ | ☐ Eukaryotic cell lines |
| ☒ | ☐ Palaeontology and archaeology |
| ☐ | ☒ Animals and other organisms |
| ☒ | ☐ Clinical data |
| ☒ | ☐ Dual use research of concern |
| ☒ | ☐ Plants |

## Methods

| n/a | Involved in the study |
|-----|----------------------|
| ☒ | ☐ ChIP-seq |
| ☒ | ☐ Flow cytometry |
| ☒ | ☐ MRI-based neuroimaging |

# Animals and other research organisms

Policy information about studies involving animals; ARRIVE guidelines recommended for reporting animal research, and Sex and Gender in Research

| Laboratory animals | Subjects were three adult male rhesus macaques (Macaca mulatta), ages 8, 11, and 12. |
|-----|-----|
| Wild animals | No wild animals were used in the study |
| Reporting on sex | As monkey electrophysiology studies typically report findings from only a few subjects (Fries & Maris, 2022), isolating effects due to gender is typically not possible. All research subjects in the present study are male. |
| Field-collected samples | No field-collected samples were used in this study. |
| Ethics oversight | All experimental procedures were approved by the Stanford University Animal Care and Use Committee and were in accordance with the policies and procedures of the National Institutes of Health. |

Note that full information on the approval of the study protocol must also be provided in the manuscript.

# Plants

| Seed stocks | n/a |
|-----|-----|
| Novel plant genotypes | n/a |
| Authentication | n/a |

