## [Peer Review File · Nature]

Manuscript Title: Intermittent Rate Coding and Cue-specific Ensembles Support Working Memory

Reviewer Comments & Author Rebuttals

Reviewer Reports on the Initial Version:

Referees' comments:

Referee #1 (Remarks to the Author):

In this manuscript, Moore and colleagues reassess the notion that spatial information held in working memory is encoded in the persistent activity of neurons in the lateral prefrontal cortex. They trained monkeys in variants of the classic oculomotor delayed response task and recorded large populations of neurons simultaneously using Neuropixel probes, giving them the opportunity to investigate trial-wise neural activity. They report that, within a trial, there are idiosyncratic fluctuations in the encoding of target information during the working memory delay and argue that these are not random variations, but sharp changes in the state of population activity. They label these states 'On' and 'Off', for when the neurons are and are not encoding the memoranda, respectively. They show that these states are observable both at the population and single neuron level, and provide some evidence that they correlate with behavior, in that better decoding found during On states correlated with faster reaction times. The authors then investigate how information might be maintained during the transient Off states, by assessing markers of functional connectivity among pairs of simultaneously recorded neurons. The motivation for this analysis is the theory that information held in working memory could be stored in short-term changes in synaptic weights rather than active neuron spiking, and this might be observable as coordinated activity when the neurons do fire. Given their large populations of neurons, the authors were able to identify a small percentage of neurons with coordinated activity that differentiated target locations, and showed that this coordination was still present during Off states. They conclude that the data support the idea that mechanisms of working memory involve functional groupings of neurons that arise from short-term synaptic plasticity in prefrontal cortex.

Overall, I think these results are important. Working memory has been studied for decades because it is believed to be a fundamental function of the lateral prefrontal cortex. However, the notion that persistent activity is the underlying neural mechanism has been increasingly questioned, and many lines of evidence, now including the decoding results of this study, suggest that the theory is incomplete at best. So far, it has been challenging to test the alternate hypothesis that memories are stored in synaptic weights because the proposed process is rapid and dynamic, impeding *ex vivo* investigation, but is also not directly related to spiking activity that can be dynamically measured *in vivo*. The idea of investigating correlation patterns to infer functional connectivity is not novel, but is used to good effect here. Although the study presents some compelling data to support the synaptic reweighting hypothesis, there remain a number of gaps that warrant further investigation, which are discussed in detail below.

1. There should be more clarity on the anatomical area(s) where the recordings are from. Figure 2A highlights a region that most atlases would consider exclusively within area 8A, but the methods say that recordings were done in FEF and Brodmann areas 9/46. The distinction is important because classic working memory functions are more associated with dorsolateral prefrontal cortex (areas 9/46), particularly because lesion studies have differentiated the functions of this area on the principal sulcus and the more posterior lateral prefrontal areas like 8A. If the recordings were done across a broader area than shown in Fig 2A, the figure should be updated to reflect this, and it would be important to know whether there are any functional differences in the reported measures between sites that are more posterior (FEF) versus more anterior (dorsolateral PFC). On the other hand, if the recording sites are primarily in area 8A, the methods should be changed and the main text should highlight that these recordings are not in 9/46, which is classically associated with working memory. Further, the term 'dorsolateral prefrontal cortex', while descriptively correct is typically used for 9/46 and it would be more appropriate to descriptively call 8A something like 'lateral prefrontal cortex', since this term doesn't have the same connotation. Finally, if the recordings are from 8A, this might slightly alter the interpretation of On and Off states. For instance, FEF appears more related to visuospatial attention (also required for this task) over working memory functions like manipulation of information, and the literature on fluctuations of attention, particularly from Kastner and colleagues, could be highly relevant.

2. I'm curious about the relationship of the present results to sequential codes that are commonly found in prefrontal cortex, often intermixed with persistent coding (e.g., see work from Stokes and Murray). On one hand, there is clearly a persistent code (Fig 6B), but this doesn't necessarily mean that there isn't sequential coding also embedded in the population activity. Could the authors assess whether there is sequential coding as well to determine whether this is relevant to their results? I do note that the delay duration in this task is variable, and modeling by Orhan and Ma (2019) suggests unpredictability could increase the prevalence of persistent over sequential activity. Since decoders in the main analyses were trained and tested on the same time points, they could rely on both persistent and sequential codes. If sequential activity is present in this task, it would be interesting to know whether results differ if decoders were trained using only persistent or only sequential components of the population activity (there are usually different units contributing to each pattern). In addition, it would be important to know the results of the CCG analyses on persistent versus sequential units.

3. I'm also curious to know more about the putative 'On' and 'Off' states:

a. The authors show that there is an overall relationship between decoding strength and reaction times. Is there any relationship between RT the state of the decoder at the time the 'go' cue appears?

b. Or does decoding/prevalence of On states predict more accurate behavioral reports on the memory guided saccade task?

c. It's a bit surprising that there isn't a relationship of states to microsaccades (Fig S4). However, those data are presented for all microsaccades, not just to the remembered location. Since the monkey is likely to microsaccade to that location and back, averaging all of these together might dilute an effect in the neural responses. Could the authors show the same information only for microsaccades in the direction of the remembered location on that trial?

- d. If decoders are trained only on Off states, is there any cue information that can be recovered, and if so, does this generalize to On states (i.e., train on Off states, test on On states)?
- e. Finally, attentional fluctuations have been linked to the phase of low frequency LFPs. The On/Off states here could be related to such fluctuations, so it's important to know whether there are there any relationships between On/Off states and LFP phases.

4. Can the authors describe the recordings in more detail:

- a. Were probes inserted perpendicular to cortex, so that they recorded a column/pseudocolumn, or was the approach oblique?
- b. Were there any differences in neurons recorded closer to the cortical surface versus deeper?
- c. Was there any anatomical organization to the functionally connected units that were found? For instance, were they close to each other or distributed? found only at certain penetration sites?
- d. Can any directionality be inferred from the CCG lags, such as superficial to deep or vice versa?
- e. Were more than one Neuropixel probe used in a session? If so, are the decoded states similar between probes or are they local? Was there ever significant cross-correlations among neurons recorded from different probes?

5. Finally, if I understand the mixture modeling that tested for distinct states in the decoding results, I believe that the model with two distributions has more free parameters but the model selection methods don't account for this. Can the authors either correct for the additional free parameters or establish a control to show that the method is not biased toward identifying two distributions? For instance, if only On state data were fit, does the method select the single-distribution model?

Minor

Can the authors clarify the temporal resolution of the decoding analyses? Presumably this was run on time bins, but I don't see the width or step of these bins clearly stated anywhere.

Referee #2 (Remarks to the Author):

This study reports results from massively parallel (neuropixels) spike recordings in the dorsolateral prefrontal cortex (DLPFC) and frontal eye fields (FEF) of awake behaving non-human primates performing a working memory (WM) task. The authors report that WM contents are not continuously maintained in the single-trial pattern of spike rates, in line with previous reports. They report instead an alternation between 'On' and 'Off' states. Furthermore, the authors analyze the patterns of functional connectivity during the memory delay period for different memoranda, and report that these patterns are memorandum-specific.

The manuscript is very well-written and the topic (persistent spiking versus rapid synaptic plasticity underlying WM maintenance) is important and timely. Furthermore, the data acquired are highly impressive; recordings of spikes in awake behaving macaques from many units at the same time, from contacts spaced closely together, have the potential to greatly increase our understanding of neural coding and its dynamics. I was happy to see the authors commit to openly releasing these valuable data upon publication.

The paper makes two basic claims, each of which can essentially be evaluated in isolation. First, that delay activity alternates between 'On' and 'Off' states; second, that functional connectivity patterns during memory delays carry memorandum-specific information. I have strong reservations with regards to the first claim, while critical information is missing to allow a proper evaluation of the second claim.

The argument for the existence of On and Off states goes as follows. The authors train a logistic regression classifier to distinguish two memory conditions (typical clock-like location stimuli for memory-guided saccade or cued target selection; the decoder discriminates a given location versus the one 180 degrees opposite) based on the neural population's firing rate pattern during the memory delay. Logistic regression outputs probability values for one stimulus class over the other, and the authors interpret these as 'classifier confidence'. They compare the observed confidence to a randomization-based null distribution, and label above-chance confidence clusters as 'On' states, while contiguous non-significant clusters are labelled 'Off' states.

The key issue here is the following. Several things could lead to worse discriminability, and hence lower 'confidence', and therefore a likely 'Off' state. One obvious candidate is a drop in firing rates, another may be a simple reduction in signal-to-noise ratio (SNR) of the recording. The authors indeed report that during Off states both firing rates reduce (back to baseline levels) and memory selectivity is lost. This is trivially true: by definition, Off states have low classifier confidence. Classifier confidence is a direct consequence of a combination of firing rate (and corresponding SNR) and memory selectivity. (Without meaning to sound overly negative, but perhaps making the issue as clear as possible: analyzing On vs Off states in terms of firing rates and memory selectivity is a case of 'double dipping'.)

The difference in rates and selectivity between putative On and Off states can therefore not be used as an argument in favour of On/Off corresponding to a true neural phenomenon. The other argument offered by the authors for On/Off identifying a true distinction is a fit of the distribution of

confidence values in the memory delay (Fig S5). This distribution is better described by a mixture model of two Beta distributions than by a single Beta distribution. This is in any case rather weak evidence for the existence of two distinct states (perhaps the shape of Beta is just not the right shape). Specifically and furthermore, (a) periods of high selectivity likely correspond to near-perfect (~ 1.0) classifier confidence, automatically leading to a peak close to 1.0 (as visible in the plot); and (b) the peak that presumably should correspond to 'Off' falls around 0.7, with no peak evident at all around 0.5, which is where a true Off state should fall, by the authors' definition.

Turning to the second claim: that PFC synaptic connectivity patterns in the memory delay reflect memory-specific information. If true, this would be huge, and the field has been waiting for a direct test of this hypothesis for a long while. At present, unfortunately, it is not possible to judge to what extent this claim is borne out by the evidence presented.

The authors only show cross-correlograms (CCGs) for a single example cell pair (Fig 5B) that happen to be distinct for two memory conditions. Several criteria are mentioned in the text that a CCG should satisfy before it counts as 'significant' (perhaps better to call it 'reflecting significant connectivity' by the way). While by themselves these criteria do not sound unreasonable, it is possible that many different CCG patterns may satisfy these criteria. To conclude synaptic connectivity from a CCG, and even more, a modulation of this connectivity, requires strong and strict evidence, such as a distinct peak only at a short time lag, and equal-height CCG tails. It is essential that the authors show many more example CCGs that satisfy their selection criteria, and also some that do not satisfy these, e.g. in a supplemental figure. More details (potentially also including further illustrations) on what does and does not count as 'connected' and 'modulated' (two distinct properties) would also help. That would allow the readers (and reviewers) to judge whether these are picking up true synaptic connectivity or something else.

Some specific notes on the connectivity analysis pipeline:

- When rate-normalizing the CCGs, take care to do this using the condition-averaged firing rate (or CCG). Otherwise you may end up shifting a high CCG in one condition to the same 'tail' level as a low CCG in another condition, while only the high CCG shows a short-latency peak. That could be due to firing rate differences, even though you normalize the rates; a strongly spiking cell (pair) will unavoidably allow you to detect peaks more readily than a silent pair. A strong test of connectivity modulation would be to focus on cell pairs that have (un-normalized!) CCGs of equal tail height in two memory conditions, but specifically differ in (only) a short-latency peak. (Note that this is a stricter criterion than what you are controlling for in Fig S6.)

- When constructing the null distribution to compare the Manhattan distances against, take care to keep data for different cells always matched. Maybe you are already doing this, but it is not explicit in the text. I.e. data for all cells that were recorded together during a particular true condition should be kept together in each permutation and assigned a single randomized pseudo-condition. If you destroy the matching of cells, that may introduce a (reducing) bias in the null distribution.

- Furthermore, relatedly, the null hypothesis per cell pair is that 0 and 180 degrees are indistinguishable, so you should only permute the 0 and 180 degrees condition when constructing

the null (not simply randomize all 8 conditions and repeat the entire pipeline).

It struck me that the authors appear careful to not explicitly present their evidence as reflecting a modulation of *effective* *synaptic* connectivity (only: 'functional connectivity' or 'neuronal ensembles'). While I understand this (given the unclarities around the CCG pipeline above), the key outstanding issue in the literature is really about rapid modulations of effective synaptic connectivity. (Which is also how the authors frame the study overall.) Therefore, I would recommend to sharpen up the evidence as outlined above and, if clear results are found, be bold and truly claim that synaptic patterns carry memory information. This would be a landmark finding that stands on its own, even if the On/Off state analysis may end up having to be removed. These should be ideal data to test the synaptic-plasticity-underlies-WM hypothesis. Weaker claims like functional connectivity patterns corresponding to memory contents have been successfully put forward in the literature already, as the authors rightly acknowledge.

Minor:

The regions recorded from (FEF/DLPFC) should be identified early in the main text (and/or even abstract).

It would strengthen the conclusions to show some key results separately for the two animals in the supplement. (Especially important with claims of two states etc.)

Define 'confidence' of the classifier (i.e. $p(x = X)$ output) early on in the main text (p. 7) – though note the issues regarding this analysis above.

Fig S1 reflects spatial distance, it would be helpful to make this explicit (even better, if possible, change the normalization such that you can show units of mm/ μ m).

Signed,
Eelke Spaak

Referee #3 (Remarks to the Author):

Working memory is often associated with “persistent activity”, which is cue-specific neural activity typically found during the delay interval of a working memory task. Here, the authors find that neurons in PFC have up and down states during delay intervals, with persistent activity completely eliminated during the down states. They then find that during the down states, some neurons show tuned pairwise correlations in spiking, which the authors suggest could mediate working memory due to short term synaptic plasticity. Although the results are novel and intriguing, the conclusions seem premature given the limitations of the data.

1. If the brain structures mediating spatial working memory all showed synchronized down states with persistent activity eliminated, it would be a puzzle how persistent activity alone could mediate working memory. Silent, but sensitized synapses could play an important role in that case, as the authors suggest. However, the authors have only shown down states in limited recordings in the FEF and adjacent prefrontal cortex. It is not clear how widespread the neuropixel recordings were in a given session – did they cover 1% of PFC or 10%? Whatever the coverage, it was unlikely to include most of the PFC with persistent activity, and the authors acknowledge that the down states might be local phenomena related to traveling waves, etc rather than global properties of large regions of cortex. Given that many other regions of PFC might have retained their persistent activity during the local down state, it seems equally possible that this persistent activity in other PFC regions entirely mediates working memory and not sensitized synapses in FEF. The existence of local down states in PFC during the delay is not strong evidence against the idea that persistent activity mediates working memory.

2. Along these lines, it is striking that a recent (2023) bioRxiv preprint from the authors’ own lab reports that the FEF is specifically not important for spatial working memory since they find that deactivation of FEF impairs saccades but not working memory. Muscimol deactivation seems like it would be the equivalent of a very large down state. If FEF is not important for spatial working memory, doesn’t this call into question the basis for the current working memory experiment? Are the authors recording in the wrong place?

3. Not only did the recordings not encompass most of the PFC with presumed persistent activity, they certainly did not encompass other structures with persistent activity in spatial working memory. Most notably, Pat Goldman-Rakic showed many years ago (2000) that deactivation of PFC did not eliminate persistent activity during WM in parietal cortex, and deactivation of parietal cortex did not eliminate persistent activity during WM in PFC, suggesting that they were independent sources of persistent activity in working memory. Couldn’t the parietal cortex hold the memory of the spatial location online during a PFC down state, and even restore the persistent activity during the up states, even if the PFC down states were global, not local?

4. The idea that persistent activity in one structure might be eliminated but then restored due to persistent activity in another structure was specifically supported by the Svoboda lab (2016), at least for premotor activity. Deactivation of persistent activity in one hemisphere was restored by persistent activity in the other hemisphere. Furthermore, the Svoboda lab showed the importance of thalamo-cortical loops for persistent activity (2017).

5. The most intriguing evidence is that some pairwise cross-correlograms of cells are tuned to the cue during the delay. One thing I don’t understand is that the authors report that only 2% of the pairs recorded show tuned evoked responses to the cue during the cue period. Why are the responses of so few cells tuned to the cue? This seems different from what previous studies have

reported, and it raises additional questions about whether this region is actually important for working memory. They also find that 1.5% of pairs showed significant CCGs to a given cue. They report that the percentage of cells tuned to the cue and participating in a CCG for the same cue during the delay is 2.5x greater than expected by chance. Fair enough, but it would be more useful to know what percentage of cells that respond selectively to a cue also participate in a CCG to the same cue during the delay. If I make a simple calculation myself, it seems like this is .075% of the cells showing this property. I accept that this is statistically significant, but it seems like an extremely small number of pairs to be the basis of claims about the basis for working memory.

6. Finally, I am not sure why the finding that some PFC pairs show cue-specific correlated activity during the delay means that their synapses must be facilitated. Couldn't the synapses be unmodified but the cells receive a subthreshold synaptic input from another source? This subthreshold input may influence the probability of joint spiking but not change firing rates. For example, maybe the thalamus or the parietal cortex targets pairs of cells in PFC that are tuned for a given cue location. I don't understand how this study localizes the source of plasticity to the synapses in the FEF.

Author Rebuttals to Initial Comments:

Referees' comments:

We thank the reviewers and the editors for their careful reading of our manuscript and their helpful comments and suggestions. We have made substantial changes to the revised manuscript and performed additional analyses to address their suggestions, questions and concerns. Most notably, we have also expanded the scope of our electrophysiological recordings with new experiments in a third monkey. The additional experimental data clearly demonstrate that our results generalize across subregions of LPFC. Furthermore, new analyses validate the rigor of both our classification and CCG-based analyses. Altogether, the data show - for the first time - lapses in mnemonic coding in LPFC on single trials. Furthermore, we show that those lapses are coupled with residual information about memoranda in the absence of a firing-rate code. The latter discovery is a key prediction of synaptic models (e.g. Mongillo et al, 2008), and necessarily constrains future mechanistic models of working memory.

We provide a point-by-point response to each of the comments below.

Referee #1 (Remarks to the Author):

Overall, I think these results are important. Working memory has been studied for decades because it is believed to be a fundamental function of the lateral prefrontal cortex. However, the notion that persistent activity is the underlying neural mechanism has been increasingly questioned, and many lines of evidence, now including the decoding results of this study, suggest that the theory is incomplete at best. So far, it has been challenging to test the alternate hypothesis that memories are stored in synaptic weights because the proposed process is rapid and dynamic, impeding *ex vivo* investigation, but is also not directly related to spiking activity that can be dynamically measured *in vivo*. The idea of investigating correlation patterns to infer functional connectivity is not novel, but is used to good effect here.

Thank you for the positive overall assessment!

Although the study presents some compelling data to support the synaptic reweighting hypothesis, there remain a number of gaps that warrant further investigation, which are discussed in detail below.

1. There should be more clarity on the anatomical area(s) where the recordings are from.

We thank the reviewer for these helpful comments and agree that more clarity on the recording sites was needed. The recordings reported in our original submission spanned both area 8, including the frontal eye field (FEF) near the anterior bank of the arcuate, and the posterior principalis (9/46), (Monkey A, more posterior, area 8; Monkey H, more anterior, area 8 & area 9/46). Recordings in area 8 were largely anterior to sites from which saccades could be evoked

with low-threshold microstimulation, which defines the FEF (Bruce et al., 1985). We now state this explicitly in the main text (lines 77-78) and methods (lines S36-37).

Figure 2A highlights a region that most atlases would consider exclusively within area 8A, but the methods say that recordings were done in FEF and Brodmann areas 9/46.

Thank you for highlighting that Fig. 2A was unclear - the original Figure 2A did include posterior 9/46 but this was difficult to make out because of the opacity of the shading. We have updated Fig 2A to make it clear that the recordings encompass 9/46 and to reflect the more anterior sites recorded in monkey 3 (see below).

The distinction is important because classic working memory functions are more associated with dorsolateral prefrontal cortex (areas 9/46), particularly because lesion studies have differentiated the functions of this area on the principal sulcus and the more posterior lateral prefrontal areas like 8A. If the recordings were done across a broader area than shown in Fig 2A, the figure should be updated to reflect this, and it would be important to know whether there are any functional differences in the reported measures between sites that are more posterior (FEF) versus more anterior (dorsolateral PFC).

Figure R1. Coronal slice at +38 mm AP EBZ, imaged at the midline of the recording chamber implanted in Monkey J, with example electrode trajectory. Recordings spanned +36-40 mm AP.

To address the reviewer's concern about possible differences across regions within LPFC, we trained a third monkey (J) to perform the oculomotor delayed response task, explicitly targeting more anterior portions of 9/46 (Fig. R1), and have integrated these results into the manuscript. Our core results are highly consistent across Monkey A (area 8), Monkey H (area 8 and posterior 9/46), and Monkey J (anterior 9/46)(Fig. S9 and Fig. R2, below).

Figure R2. Top: Single-animal memory tuning functions for held-out units during On and Off states. Tuning functions show the mean normalized firing rate during the memory delay (z-scored across trials) for held-out units, relative to each unit's preferred cue location. Error bars (small) denote SEM. Bottom: Single-animal mean normalized Manhattan distance, using data from the entire memory delay (gray), only during On states (orange), and only during Off states. Violin plots show bootstrap across sessions. Asterisks denote significance (* $p < 0.05$; ** $p < 0.01$), 'ns', nonsignificant.

Thus, as we now describe in the main paper, our recordings spanned areas 8 and a large swath of 9/46, using the designations of Petrides, 2005. Notably, this heavily overlaps with most of the past neurophysiological studies reporting spatial delay period activity within LPFC (see Fig. R3 below).

Figure R3. Examples of prior summaries of recording areas of LPFC in studies of persistent spatial working memory activity. Areas within LPFC largely involved recording sites within the posterior part of sulcus principalis (dorsal or ventral) and area 8A. From: (A) Funahashi et al., 1993 (B) Rainer et al., 1998 (C) Constantinidis et al., 2018 (D) Bastos et al., 2018 (E) Busch et al., 2024 (F) Chafee & Goldman-Rakic, 2000 (G) Lundqvist et al., 2016

On the other hand, if the recording sites are primarily in area 8A, the methods should be changed and the main text should highlight that these recordings are not in 9/46, which is classically associated with working memory. Further, the term ‘dorsolateral prefrontal cortex’, while descriptively correct is typically used for 9/46 and it would be more appropriate to descriptively call 8A something like ‘lateral prefrontal cortex’, since this term doesn’t have the same connotation.

We agree the term LPFC is less confusing and have updated the text accordingly.

Finally, if the recordings are from 8A, this might slightly alter the interpretation of On and Off states. For instance, FEF appears more related to visuospatial attention (also required for this task) over working memory functions like manipulation of information, and the literature on fluctuations of attention, particularly from Kastner and colleagues, could be highly relevant.

We agree that the relation between these results and the work on rhythmic attention from Kastner and colleagues should be made clear. The dynamics of the On/Off states and their relationship to the local field potential suggest a distinct phenomenon. We unpack this more in response to comment 3e, below.

2. I’m curious about the relationship of the present results to sequential codes that are commonly found in prefrontal cortex, often intermixed with persistent coding (e.g., see work

from Stokes and Murray). On one hand, there is clearly a persistent code (Fig 6B), but this doesn't necessarily mean that there isn't sequential coding also embedded in the population activity. Could the authors assess whether there is sequential coding as well to determine whether this is relevant to their results? I do note that the delay duration in this task is variable, and modeling by Orhan and Ma (2019) suggests unpredictability could increase the prevalence of persistent over sequential activity. Since decoders in the main analyses were trained and tested on the same time points, they could rely on both persistent and sequential codes. If sequential activity is present in this task, it would be interesting to know whether results differ if decoders were trained using only persistent or only sequential components of the population activity (there are usually different units contributing to each pattern). In addition, it would be important to know the results of the CCG analyses on persistent versus sequential units.

We agree that this is a good question, and we have revised the paper and analyses to explore it further. In this dataset, we observe largely stable (rather than time-varying or sequential) trial-averaged mnemonic coding during the memory delay. As the reviewer notes, the stable code is evident from the strong cross-temporal classification performance in Fig. 6B. To explicitly quantify the extent to which mnemonic coding during the memory delay was stable or sequential, we used demixed principal components analysis (dPCA; Kobak et al., 2016) to quantify how much variance in neural population state during the memory delay was explained by cue location, time, and a cue by time interaction. A stable population code would manifest as a large effect of cue; sequential or other time-varying codes would result in a large interaction term. Across sessions, cue explained 29.3% of population variance, while the interaction of cue and time explained 4.5% (Fig. S12a, Fig. R4a, below). Time-varying signals unrelated to mnemonic coding explained 9.3% of population variance. Accordingly, stable coding was readily apparent in the largest dPCA components (Fig. S12b-c, Fig R4b-c).

An analysis of single neurons yielded similar results. For each neuron, we used ANOVA to model trial-averaged firing rates during the memory delay as a linear combination of cue location, time, and their interaction, and then identified the proportion of neurons for which each term was significant (Fig. S12d, R4d). We observed a significant effect of cue in 67% of neurons, time in 29%, and their interaction in 7.6% (an average of 25 neurons per session). Together, these results suggest that time-varying mnemonic coding is not a strong feature of these data and also highlight practical issues of isolating and studying the few neurons with time-varying coding of cue (since both our classification and CCG analyses depend on large numbers of simultaneously recorded neurons). These results are consistent with previous work (Murray et al., 2017) suggesting that stable population mnemonic coexist with cue-invariant temporal dynamics in the prefrontal cortex.

More generally, the reviewer's suggestion to consider the possibility that a second, 'minor' population code (sequential or otherwise) may be spanning putative 'Off' states is very well-taken, and we believe their analysis proposed in comment 3d directly addresses this issue (see below).

A**D****B****C**
Fig R4. (A) Proportion of variance explained by each of the task components (dPCA). Violin plots reflect bootstrap across sessions. (B) Stable coding was also evident in the demixed components identified by dPCA. As with conventional PCA, each of these components can be understood as a linear combination (or, critically, a linear readout) of the population of trial-averaged single-neuron PSTHs. For 21 of the 25 total experimental sessions, the dPCA component explaining the most variance in the data encoded the cue in a time-invariant manner. The largest component from the remaining 4 sessions was a cue-invariant ramp unrelated to mnemonic coding; for these sessions (C), the second component captured time-invariant information about the cue. Consistent with the analysis of explained variance, none of these components displayed sequential coding or other time-varying signals. (D) Proportion of neurons displaying a significant effect of cue, time, and their interaction during the memory delay.

3. I'm also curious to know more about the putative 'On' and 'Off' states:

a. The authors show that there is an overall relationship between decoding strength and reaction times. Is there any relationship between RT the state of the decoder at the time the 'go' cue appears?

Thanks for this suggestion - we used regression to examine the relationship between the state of the decoder ('On' or 'Off') at the go cue and reaction times while controlling for task and cue location. In all three animals, 'On' states were associated with faster reaction times (Fig. R5). We have integrated these results into the manuscript (lines 191-192, Fig. S2C).

Fig. R5. Regression coefficient relating state (On/Off) immediately after fixation offset to reaction time in milliseconds. Negative values indicate that On states are associated with reduced reaction times. Violin plots show bootstrap across sessions.

b. Or does decoding/prevalence of On states predict more accurate behavioral reports on the memory guided saccade task?

This is a great suggestion. Indeed, state at go cue predicted the accuracy of behavioral reports in all three monkeys (Fig R6, below). We have integrated these results into the manuscript (Fig. S2B).

Fig. R6. Behavioral performance of the animal (proportion correct) for trials in which confidence was 'On' (orange) or 'Off' (blue) immediately after the go cue (fixation offset). Error bars are 95% CI (binomial).

c. It's a bit surprising that there isn't a relationship of states to microsaccades (Fig S4). However, those data are presented for all microsaccades, not just to the remembered location. Since the monkey is likely to microsaccade to that location and back, averaging all of these together might dilute an effect in the neural responses. Could the authors show the same information only for microsaccades in the direction of the remembered location on that trial?

We agree that this is an important additional control. The animals were indeed slightly more likely to make microsaccades toward the previously cued location (Fig. R7c, below). However, rerunning this analysis on only those microsaccades towards the previously cued location (+/- 20 degrees) yielded similar results (Fig. R7d). We have integrated these results into the manuscript (Fig. S4).

Figure R7. Classifier confidence was not affected by microsaccades. (A) Example trial showing eye speed data and microsaccade identification. Microsaccades were identified as peaks in eye speed >10 DVA/s. (B) Mean classifier confidence during the memory delay, locked to microsaccades (average across $N = 8,910$ microsaccades). Shaded area (small) indicates 95% confidence intervals. (C) Probability density of microsaccade angles during the memory delay, relative to the cued location. (D) Mean classifier confidence during the memory delay, locked to microsaccades to the cued location (± 20 degrees).

d. If decoders are trained only on Off states, is there any cue information that can be recovered, and if so, does this generalize to On states (i.e., train on Off states, test on On states)?

This is a good suggestion. As outlined by the reviewer, we have repeated our classification analysis for each trial and time point during the memory delay (500-1400 ms postcue), this time only training on trials with Off states (or, separately, On states) at the same time point as test. Across sessions, confidence was significantly above chance only when training on On states and testing on a held-out On state (Fig. R8, below). These results indicate that Off states reflect time periods with no reliable information, and not a less frequent, second coding scheme that our original classifier failed to capture. We now report these results on lines 192-195 and Fig. S6.

Figure R8. Cross-validated classifier confidence, binned by training and test state. Left: classifier confidence when trained on Off states and tested on Off and On states. Right: classifier confidence when trained on On states and tested on Off states. Violin plots show bootstrap across sessions.

e. Finally, attentional fluctuations have been linked to the phase of low frequency LFPs. The On/Off states here could be related to such fluctuations, so it's important to know whether there are there any relationships between On/Off states and LFP phases.

Thanks for this suggestion. We have run the analysis described by the reviewer - in brief, we tested if the phase of a range of frequency bands was predictive of the onset of On and Off states. We were not able to identify any significant phase-state relationships (Figure R9, Fig S7). This suggests that the fluctuations in confidence we observe are distinct from the rhythmic sampling of attention described by Fiebelkorn et al., 2018, and others. These results are also consistent with the lack of clear periodicity in single trial confidence (Fig 2) and the cognitive processes under study here (i.e., maintaining a single location in memory vs covertly monitoring

several locations for the appearance of a target). We now describe these results in the main text (lines 195-198).

Fig. R9. Mean phase-state relationship (z-scored) as a function of frequency. Gray traces show individual sessions (N=25). Values reflect absolute angular difference between the mean phase of On and Off states, normalized by the null (randomization test).

4. Can the authors describe the recordings in more detail:

a. Were probes inserted perpendicular to cortex, so that they recorded a column/pseudocolumn, or was the approach oblique?

We have clarified these details in the revised paper. Recordings from the probe appeared to span several cortical columns, as inferred from the broad distribution of preferred cue locations across neurons. For example, Figure R10, below, shows a particularly clear progression in the preferred cue location along the length of the probe. We now note this important detail in the methods (lines S40-41).

Fig. R10. Preferred cue location (angle relative to the contralateral horizontal meridian) of selective neurons as a function of distance from probe tip (example session).

b. Were there any differences in neurons recorded closer to the cortical surface versus deeper?

Interestingly, we didn't find any relationship between depth and different functional subclasses, though we interpret this with caution because of our non-perpendicular approach. Nevertheless, readers will likely have the same question so we now note this in our description of the clustering result, with appropriate caveats (Fig S1).

c. Was there any anatomical organization to the functionally connected units that were found? For instance, were they close to each other or distributed? found only at certain penetration sites?

Yes - consistent with previous work (Trepka et al., 2022), probability of functional connection decreased with spatial distance (Fig R11):

Fig. R11. Probability of two units displaying a functional connection as a function of their distance from each other along the probe. Dots reflect mean across sessions.

Additionally, functionally connected units were found at all penetration sites. We now note these details in the manuscript (lines 251-253).

d. Can any directionality be inferred from the CCG lags, such as superficial to deep or vice versa?

Indeed, among pairs with a significant functional connection, the deeper neuron tends to spike before the more superficial neuron (Fig. R13, below). We agree that this and similar attempts to infer circuit architecture from functional connectivity are extremely interesting and have begun to

pursue such questions in detail in visual cortex (e.g., Trepka et al., 2022) and are in the process of applying similar approaches to these data. However, we believe such questions are beyond the scope of the present work, as our main finding here is that there is information about remembered stimuli in patterns of functional connectivity.

Fig. R13. Mean peak lag (ms) as a function of the signed distance between each significantly connected pair. Negative distance indicates that the neuron under consideration was deeper than its pair mate; negative lag indicates that the neuron under consideration tended to spike first.

e. Were more than one Neuropixel probe used in a session? If so, are the decoded states similar between probes or are they local? Was there ever significant cross-correlations among neurons recorded from different probes?

Thanks for drawing this omission to our attention. We used one Neuropixel probe in each session; we now clarify this in the methods (line S38-39). We agree that studying the spatial heterogeneity of these dynamics is an important future direction and experiments to that end are imminent. Nevertheless, single probe recordings are sufficient to address our question here, given that the canonical accounts of working memory currently in textbooks predict persistent activity within local populations of neurons of the sort that we measure here.

5. Finally, if I understand the mixture modeling that tested for distinct states in the decoding results, I believe that the model with two distributions has more free parameters but the model selection methods don't account for this. Can the authors either correct for the additional free parameters or establish a control to show that the method is not biased toward identifying two distributions? For instance, if only On state data were fit, does the method select the single-distribution model?

Thanks for drawing our attention to this omission - the mixture model analysis is cross-validated and so controls for the difference in the number of free parameters of the competing models. We now emphasize this in Fig. S5 and lines 168-169.

Minor

Can the authors clarify the temporal resolution of the decoding analyses? Presumably this was run on time bins, but I don't see the width or step of these bins clearly stated anywhere.

Yes - thanks for pointing this out. For the decoding analyses, spike trains were smoothed with a 100-ms boxcar and analyzed in 10 ms steps (lines S56-58).

Referee #2 (Remarks to the Author):

This study reports results from massively parallel (neuropixels) spike recordings in the dorsolateral prefrontal cortex (DLPFC) and frontal eye fields (FEF) of awake behaving non-human primates performing a working memory (WM) task. The authors report that WM contents are not continuously maintained in the single-trial pattern of spike rates, in line with previous reports. They report instead an alternation between 'On' and 'Off' states. Furthermore, the authors analyze the patterns of functional connectivity during the memory delay period for different memoranda, and report that these patterns are memorandum-specific.

The manuscript is very well-written and the topic (persistent spiking versus rapid synaptic plasticity underlying WM maintenance) is important and timely. Furthermore, the data acquired are highly impressive; recordings of spikes in awake behaving macaques from many units at the same time, from contacts spaced closely together, have the potential to greatly increase our understanding of neural coding and its dynamics. I was happy to see the authors commit to openly releasing these valuable data upon publication.

We thank the reviewer for these positive comments.

The paper makes two basic claims, each of which can essentially be evaluated in isolation. First, that delay activity alternates between 'On' and 'Off' states; second, that functional connectivity patterns during memory delays carry memorandum-specific information. I have strong reservations with regards to the first claim, while critical information is missing to allow a proper evaluation of the second claim.

Below, we directly address the concerns regarding both of the claims.

1) The argument for the existence of On and Off states goes as follows. The authors train a logistic regression classifier to distinguish two memory conditions (typical clock-like location stimuli for memory-guided saccade or cued target selection; the decoder discriminates a given location versus the one 180 degrees opposite) based on the neural population's firing rate pattern during the memory delay. Logistic regression outputs probability values for one stimulus class over the other, and the authors interpret these as 'classifier confidence'. They compare the observed confidence to a randomization-based null distribution, and label above-chance confidence clusters as 'On' states, while contiguous non-significant clusters are labelled 'Off' states.

The key issue here is the following. Several things could lead to worse discriminability, and hence lower 'confidence', and therefore a likely 'Off' state. One obvious candidate is a drop in firing rates, another may be a simple reduction in signal-to-noise ratio (SNR) of the recording.

The reviewer makes an excellent point about an alternative interpretation of the On and Off states. In our original submission, we assumed that firing rates were lower during Off states due to a decrease in (biological) signal. However, the reduced firing rates we observed during Off states could equally be a product of an increase in (recording) noise. Therefore, we compared the standard deviation of the background noise (Donoho & Johnstone, 1994; Quiroga et al., 2004) during On and Off states. In brief, we measured the median absolute deviation of the raw high-pass filtered data during On and Off states. Because the median is less contaminated by sparse, high-amplitude values driven by spiking, this metric provides a more robust estimate of recording noise SD than mean squared deviation (Quiroga et al., 2004). Importantly, background noise did not significantly differ across On and Off states (Figure R14, $p = 0.339$); indeed, the maximum difference in noise level across all sessions was a fraction of a microvolt ($0.465 \mu\text{V}$). We have added a description of this analysis to the main text (lines 198-200), methods (lines S179-186), and supplementary materials (Fig. S8).

Fig. R14. Standard deviation of the background noise, in microvolts, averaged across all On and Off states. Small circles: individual session means; large circles: grand means. Background noise did not differ across On and Off states ($p = 0.339$, sign-rank).

2) The authors indeed report that during Off states both firing rates reduce (back to baseline levels) and memory selectivity is lost. This is trivially true: by definition, Off states have low classifier confidence. Classifier confidence is a direct consequence of a combination of firing rate (and corresponding SNR) and memory selectivity. (Without meaning to sound overly negative, but perhaps making the issue as clear as possible: analyzing On vs Off states in terms of firing rates and memory selectivity is a case of 'double dipping'.) The difference in rates and selectivity between putative On and Off states can therefore not be used as an argument in favour of On/Off corresponding to a true neural phenomenon.

We agree with the reviewer that it is critically important to avoid circularity when validating putative On and Off states. Indeed, to avoid this, the firing rate effects the reviewer refers to here were assessed using cross-validation. Time epochs corresponding to On and Off states were identified by applying our classification procedure to one-half of the neurons from each session; firing rate differences across the two states were identified by examining the held-out half of neurons (entirely unseen by the classifier). Exactly as the reviewer notes, without this

cross-validation procedure reduced memory selectivity during Off states would be trivially true and could be driven by chance fluctuations in neuronal firing rates (e.g., due to Poisson variability in spiking). However, the fact that On and Off states identified using one subset of neurons strongly predict the tuning of entirely ‘unseen’ neurons indicates that these states are (1) synchronized across the recorded population and (2) are not due to chance fluctuations in the firing rates of the neurons entering the classifier. We have updated both the text of the manuscript (lines 210-212) and Fig. 4D to emphasize these details and avoid confusion.

3) The other argument offered by the authors for On/Off identifying a true distinction is a fit of the distribution of confidence values in the memory delay (Fig S5). This distribution is better described by a mixture model of two Beta distributions than by a single Beta distribution. This is in any case rather weak evidence for the existence of two distinct states (perhaps the shape of Beta is just not the right shape).

Note that our choice of the beta distribution here is principled: it is extremely flexible, able to display a broad range of skewness and kurtosis, and naturally accommodates bounded continuous variables of the sort we analyze here (Verkuilen & Smithson, 2012). By intention, this flexibility makes this analysis conservative, ensuring that the 1-state model is capable of describing a broad range of empirical distributions. In this respect, the shape of the Beta is rather optimal. We now emphasize these details in the text (lines S114-119).

To provide further evidence of the existence of two states, we confirmed that the superior performance of the 2-state model is not dependent on the choice of the Beta distribution. The 2-state model also outperforms the one-state model when employing mixtures of Gaussians. The 2-state log-likelihood was larger than the 1-state log-likelihood for each of the 25 sessions ($p < 0.001$, chi-squared).

More generally, the use of model comparison to estimate the number of discrete states underlying continuous data is of course standard in the literature (e.g., Ashwood et al., 2022), and the strength of evidence of a given number of states is presumably proportional to the statistical evidence. In our case, in statistical terms, the strength of evidence for 2-states for both Beta and Gaussian mixture models is categorically “large” (Cohen’s $D = 0.8$ and 1.1 for Beta and Gaussian, respectively).

Finally, all of these results are corroborated by the data-driven, non-parametric approach for identifying and validating On and Off states that we pursue in the main text. This overall result has already begun to receive converging support from unpublished work using other approaches (Tao & Libedinsky, 2024).

4) Specifically and furthermore, (a) periods of high selectivity likely correspond to near-perfect (~ 1.0) classifier confidence, automatically leading to a peak close to 1.0 (as visible in the plot); and (b) the peak that presumably should correspond to ‘Off’ falls around 0.7, with no peak evident at all around 0.5, which is where a true Off state should fall, by the authors’ definition.

Apologies for the confusion - note that the distribution shown in Fig S5 is just a particular example from one condition in one session, with latent component modes at 0.63 and 0.93. We've added a few more examples with the latent components plotted to make this clearer (Figure R15 and Fig S5). Overall, the average modal confidence values for the On and Off distributions were 0.66 (\pm 0.05) and 0.53 (\pm 0.07), more in line with the reviewer's predictions. Note that we observed similar values for On and Off states defined non-parametrically (see Figure R8, above) and that these 'On' confidence values correspond to strong firing rate selectivity (Fig. 4).

Fig R15. (A) Histogram of confidence values during memory delay time points (one cue location) from three example sessions. Dashed line shows the best fitting beta distribution. Solid line shows the best fitting mixture of two beta distributions. Purple and yellow lines show the individual component distributions for the two-state model. (B) Histogram showing average proportion-weighted component modes (bootstrap). (C) Cross-validated model comparison results for 2-state vs 1-state fits for all sessions ($N=25$). The 2-state model outperforms the 1-state ($p=0.009$, chi-squared). Light circles show individual session scores; dark circle shows mean across sessions; black line shows equivalent model performance.

5) Turning to the second claim: that PFC synaptic connectivity patterns in the memory delay reflect memory-specific information. If true, this would be huge, and the field has been waiting for a direct test of this hypothesis for a long while. At present, unfortunately, it is not possible to judge to what extent this claim is borne out by the evidence presented.

We agree with the reviewer that this result is important, and therefore should be assessed as thoroughly as possible.

The authors only show cross-correlograms (CCGs) for a single example cell pair (Fig 5B) that happen to be distinct for two memory conditions. Several criteria are mentioned in the text that a CCG should satisfy before it counts as ‘significant’ (perhaps better to call it ‘reflecting significant connectivity’ by the way).

We agree ‘significant CCG’ is unclear and now precisely define this shorthand when first introducing this term (line 250).

While by themselves these criteria do not sound unreasonable, it is possible that many different CCG patterns may satisfy these criteria. To conclude synaptic connectivity from a CCG, and even more, a modulation of this connectivity, requires strong and strict evidence, such as a distinct peak only at a short time lag, and equal-height CCG tails.

Apologies for the confusion - we now clarify in the main text (line 248) that we consider only short time lag peaks (≤ 10 ms) as the reviewer suggests. This criterion was selected based on previous work (Siegle et al., 2021) and the empirically observed latencies of monosynaptic connections in the prefrontal cortex (González-Burgos et al., 2000). Nonetheless, note that our analyses are robust even when considering very short latencies (Fig. R16, below), such as those used by Buzsaki and colleagues to identify “putative” monosynaptic connections (e.g. Senzai et al., 2019).

Figure R16. Mean normalized Manhattan distance, as in Figure 4f, but only considering CCG lags between 1 and 3 ms and only comparing CCGs across conditions with equal-height tails.

Note also that our CCG analysis pipeline is based on established methods in the literature (Csicsvari et al., 1998; Siegle et al., 2021; Smith & Kohn, 2008) that employ both (1) firing rate normalization and (2) the subtraction of a spike-jittered ‘control’ CCG to remove spurious

correlations due to firing rate. We now emphasize the jittering procedure in the main text. We were unable to find any examples of the ‘tail-height’ matching procedure that the reviewer proposes in the literature. However, to be conservative, we ran this control and observed near-identical results when only comparing CCGs across conditions with equal unnormalized tails (Z -difference < 2 , Figure R16, above).

It is essential that the authors show many more example CCGs that satisfy their selection criteria, and also some that do not satisfy these, e.g. in a supplemental figure. More details (potentially also including further illustrations) on what does and does not count as ‘connected’ and ‘modulated’ (two distinct properties) would also help. That would allow the readers (and reviewers) to judge whether these are picking up true synaptic connectivity or something else.

Thanks for this suggestion - we agree that including many example CCGs will be helpful for readers. We now include all of the significant CCGs for two cue conditions from an example session in the supplement and an equivalent number of non-significant CCGs from each condition (Fig S10, Fig R17, below).

Figure R17. All significant CCGs from the “left” (A, pink) and “right” (B, green) cue conditions from an example session, along with an equal number of randomly sampled non-significant CCGs

(gray). Significant CCGs are arranged by the lag of their suprathreshold peak, non-significant CCGs are ordered by the lag of their maximum value.

Some specific notes on the connectivity analysis pipeline:

- When rate-normalizing the CCGs, take care to do this using the condition-averaged firing rate (or CCG). Otherwise you may end up shifting a high CCG in one condition to the same 'tail' level as a low CCG in another condition, while only the high CCG shows a short-latency peak. That could be due to firing rate differences, even though you normalize the rates; a strongly spiking cell (pair) will unavoidably allow you to detect peaks more readily than a silent pair. A strong test of connectivity modulation would be to focus on cell pairs that have (un-normalized!) CCGs of equal tail height in two memory conditions, but specifically differ in (only) a short-latency peak. (Note that this is a stricter criterion than what you are controlling for in Fig S6.)

Indeed, CCGs are normalized using the condition averaged firing rate. We now emphasize this in the text (line S204-205). Note that we describe the results of the tail-height matching control above (Fig. R16) and our results are unchanged.

- When constructing the null distribution to compare the Manhattan distances against, take care to keep data for different cells always matched. Maybe you are already doing this, but it is not explicit in the text. I.e. data for all cells that were recorded together during a particular true condition should be kept together in each permutation and assigned a single randomized pseudo-condition. If you destroy the matching of cells, that may introduce a (reducing) bias in the null distribution.

Indeed, CCGs are always shuffled within cell pair. We now emphasize this in the text (line S241).

- Furthermore, relatedly, the null hypothesis per cell pair is that 0 and 180 degrees are indistinguishable, so you should only permute the 0 and 180 degrees condition when constructing the null (not simply randomize all 8 conditions and repeat the entire pipeline).

Indeed, CCGs are only permuted across the pair of conditions under consideration. We now emphasize this in the text (line S241).

It struck me that the authors appear careful to not explicitly present their evidence as reflecting a modulation of *effective* *synaptic* connectivity (only: 'functional connectivity' or 'neuronal ensembles'). While I understand this (given the unclarities around the CCG pipeline above), the key outstanding issue in the literature is really about rapid modulations of effective synaptic connectivity. (Which is also how the authors frame the study overall.) Therefore, I would recommend to sharpen up the evidence as outlined above and, if clear results are found, be bold and truly claim that synaptic patterns carry memory information. This would be a landmark

finding that stands on its own, even if the On/Off state analysis may end up having to be removed. These should be ideal data to test the synaptic-plasticity-underlies-WM hypothesis. Weaker claims like functional connectivity patterns corresponding to memory contents have been successfully put forward in the literature already, as the authors rightly acknowledge.

We appreciate the reviewer's support for the importance of these results. Indeed, as described above, we have sharpened up the CCG evidence specifically as requested. This evidence provides strong support for cue-specific neuronal assemblies, which is a key prediction of synaptic models of working memory. Importantly, we identify these assemblies during the memory delay period, even when firing rates fail to provide information about memoranda, which is clearly distinct from previous studies making weaker claims (e.g., Barbosa et al., 2020). Nonetheless, we believe it may be best to use the more standard language to describe our measurements of putative synaptic connections, namely "functional connectivity" (e.g., Alonso & Martinez, 1998; Jia et al., 2013; Senzai et al., 2019; Siegle et al., 2021) while at the same time not shying away from the unprecedented nature of our results and their implications in the discussion.

Minor:

The regions recorded from (FEF/DLPFC) should be identified early in the main text (and/or even abstract).

Thanks for this suggestion; we now specify our recording locations on line 78.

It would strengthen the conclusions to show some key results separately for the two animals in the supplement. (Especially important with claims of two states etc.)

We agree and now show the key results for all three animals in the supplement (Fig. S3 & S9, Fig, R18, below).

Figure R18. Top: Single-animal memory tuning functions for held-out units during On and Off states. Tuning functions show the mean normalized firing rate during the memory delay (z-scored across trials) for held-out units, relative to each unit's preferred cue location. Error bars (small) denote SEM. Bottom: Single-animal mean normalized Manhattan distance, using data from the entire memory delay (gray), only during On states (orange), and only during Off states. Violin plots show bootstrap across sessions. Asterisks and 'ns' denote significance.

Define 'confidence' of the classifier (i.e. $p(x = X)$ output) early on in the main text (p. 7) – though note the issues regarding this analysis above.

Thanks - we now define confidence on lines 148-149.

Fig S1 reflects spatial distance, it would be helpful to make this explicit (even better, if possible, change the normalization such that you can show units of mm/um).

We now clarify that Fig S1 reflects spatial distance.

Signed,
Eelke Spaak

Referee #3 (Remarks to the Author):

Working memory is often associated with “persistent activity”, which is cue-specific neural activity typically found during the delay interval of a working memory task. Here, the authors find that neurons in PFC have up and down states during delay intervals, with persistent activity completely eliminated during the down states. They then find that during the down states, some neurons show tuned pairwise correlations in spiking, which the authors suggest could mediate working memory due to short term synaptic plasticity. Although the results are novel and intriguing, the conclusions seem premature given the limitations of the data.

We thank the reviewer for their positive comments regarding the results. Below, we describe how we have directly addressed the concerns regarding the conclusions we draw.

1. If the brain structures mediating spatial working memory all showed synchronized down states with persistent activity eliminated, it would be a puzzle how persistent activity alone could mediate working memory. Silent, but sensitized synapses could play an important role in that case, as the authors suggest. However, the authors have only shown down states in limited recordings in the FEF and adjacent prefrontal cortex. It is not clear how widespread the neuropixel recordings were in a given session – did they cover 1% of PFC or 10%? Whatever the coverage, it was unlikely to include most of the PFC with persistent activity, and the authors acknowledge that the down states might be local phenomena related to traveling waves, etc rather than global properties of large regions of cortex. Given that many other regions of PFC might have retained their persistent activity during the local down state, it seems equally possible that this persistent activity in other PFC regions entirely mediates working memory and not sensitized synapses in FEF. The existence of local down states in PFC during the delay is not strong evidence against the idea that persistent activity mediates working memory.

The reviewer raises an interesting question, namely to what extent does our demonstration of down states in persistent activity address the role, or lack thereof, of LPFC persistent activity in spatial working memory, particularly given that LPFC extends over a broad region? Given that our initial data set focused largely on posterior LPFC (mostly area 8 and posterior 9/46), we took this question very much to heart, particularly as it dovetailed with a related question raised by Reviewer 1 (comment #1). First, to address the question, we have added data from an additional animal in which more anterior recordings were carried out using the same task and analyses. As can be seen in the corresponding MR images (Fig. R19, below), those recordings were clearly centered in area 9/46. Thus, across the 3 animals, the breadth of our sampled areas brackets areas 8 posteriorly to 9/46 anteriorly. Crucially, this overall coverage is very similar to most of the past studies reporting on spatial delay activity in prefrontal cortex (Fig R20), and certainly more than 10% of the extent of those areas. As can be seen in the revised results, the pattern of effects within the additional animal is essentially identical to those in the first two. Moreover, the additional results strengthen the results showing cue-specific patterns of functional connectivity during Off periods.

Figure R19. Coronal slice at +38 mm AP EBZ, imaged at the midline of the recording chamber implanted in Monkey J, with example electrode trajectory. Recordings spanned +36-40 mm AP.

Second, the reviewer raises the question of what areas of cortex are needed to mediate working memory? Again, we agree that this is an important question. The implication here though is that with a loss of persistent activity in only a subset of LPFC, a remaining subpopulation of LPFC neurons will be sufficient to support spatial working memory. However, that is completely contrary to what local inactivation experiments have repeatedly demonstrated from multiple laboratories, including our own. That is, localized pharmacological inactivation of discrete sites (e.g. 1 μ L muscimol) within the areas we studied (including 9/46) is sufficient to produce a marked loss in an animal's ability to perform standard working memory tasks (Clark et al., 2014; Sawaguchi & Iba, 2001; Suzuki & Gottlieb, 2013). These deficits reveal a clear loss of performance at spatial corresponding areas of visual space. In other words, loss of activity within very small volumes of LPFC is sufficient to eliminate normal behavior at spatially specific locations. This fact should not be surprising given that there is evidence of at least some spatial topography within areas 8 and 9/46, e.g. Sawaguchi and Iba, 2001 for the latter area. That is, when a component of that spatial topography is inactivated, the animal is unable to perform the working memory task at the corresponding location (hence the classic description of such deficits as "mnemonic scotomas" by Goldman-Rakic and colleagues, Funahashi et al., 1993). Those results indeed provide "strong" evidence of a role of locally inactivated areas (8 and 9/46) in mediating working memory, and it has not been necessary to inactivate all, or even much, of LPFC to establish this fact. Notably, our results demonstrate that within the same sites within these areas, large populations of neurons undergo coordinated, intermittent losses and recovery of persistent activity. It thus follows that some mechanism other than persistent activity must be contributing to the function of these sites when that persistence is lost.

Figure R20. Examples of prior summaries of recording areas of LPFC in studies of persistent spatial working memory activity. Areas within LPFC largely involved recording sites within the posterior part of sulcus principalis (dorsal or ventral) and area 8A. From: (A) Funahashi et al., 1993 (B) Rainer et al., 1998 (C) Constantinidis et al., 2018 (D) Bastos et al., 2018 (E) Busch et al., 2024 (F) Chafee & Goldman-Rakic, 2000 (G) Lundqvist et al., 2016

Furthermore, although other areas of the brain, such as parietal cortex, have been shown to exhibit clear spatial delay activity, inactivation of sites within LPFC is clearly more potent than inactivation of those areas. This is evident from two previous studies, the first is Chafee and Goldman-Rakic, 2000. They indeed show that delay activity in parietal and prefrontal cortex appear to be interdependent, i.e. are both affected by inactivation of the other area. However, they also show that the behavioral effects are considerably larger after inactivating prefrontal cortex (Fig. R21). Note also that the part of LPFC focused on in that study was area 8a (i.e. posterior PFC)(Fig. R20F).

Figure R21. C: Saccade errors observed when prefrontal cortex was cooled (dark gray) exceed those when the prefrontal cortex was at normal temperature (light gray). D: saccade errors when parietal cortex was cooled (dark gray) did not differ significantly from those observed when parietal cortex was at normal temperature (light gray). (from Chafee and Goldman-Rakic, 2000)

Similar observations were made by Suzuki and Gottlieb, 2013, who compared parietal and prefrontal delay activity and the impact of localized intracortical inactivation. In that study, the area of LPFC targeted was “the posterior portion of the principal sulcus just anterior to the pre-arcuate region from which saccades are elicited with low-threshold microstimulation (FEF)”. As with Chafee and Goldman-Rakic, they found that the effects of LPFC inactivation were much larger than those with parietal (LIP) inactivation (their Figure 6). Note that this was the case in spite of the fact that the volume of muscimol injected for LPFC inactivation (1 μ L) was between 3x and 8x *smaller* than that used in parietal cortex (see methods)! More importantly, the volume of pharmacological inactivation used in LPFC is on the same scale (mm) as the local populations of neurons we recorded simultaneously from in this study.

The above evidence means that we needn't speculate on the amount of LPFC that mediates spatial working memory. It has been shown that even local populations roughly on the same scale as each of our recordings is necessary for spatial working memory performance at corresponding spatial locations. Accordingly, the On/Off states observed in these local populations reliably predict the animals' accuracy and reaction time (Fig S2). Again, this fact makes sense given evidence of an apparent topography within area 8 and 9/46, and the corresponding spatially localized deficits. (As an extreme analogy, if we were reporting evidence that area V1 neurons exhibit synchronous intermittent [On/Off] encoding of orientation selectivity, such evidence would be strong even if we had only sampled a portion of the lower visual field representation. This is true because of course we know that inactivation of that representation largely eliminates vision -and certainly orientation discrimination - within a part of visual space. It would not be necessary to study the whole V1 representation to draw conclusions about the significance of the On/Off states.)

2. Along these lines, it is striking that a recent (2023) bioRxiv preprint from the authors' own lab reports that the FEF is specifically not important for spatial working memory since they find that deactivation of FEF impairs saccades but not working memory. Muscimol deactivation seems like it would be the equivalent of a very large down state. If FEF is not important for spatial working memory, doesn't this call into question the basis for the current working memory experiment? Are the authors recording in the wrong place?

We thank the reviewer for the opportunity to clarify this point. As we indicate above, and more clearly in the revised paper, our recordings extended well beyond the FEF, which is defined only as the area from which saccades can be evoked with $<50\mu\text{A}$ currents (Bruce et al., 1985). The paper referred to (now Jonikaitis et al., 2023, J Neurosci.) shows that inactivation restricted to the FEF results in profound deficits in the classic spatial working memory tasks, but that a control task suggests that such deficits are more related to motor planning than non-effector-specific spatial working memory. Other evidence suggests that more anterior LPFC, which comprises much of our data in the present paper, contains a more abstract spatial working memory signal (Funahashi, Chafee, et al., 1993), compared to the FEF or subcortical structures with delay activity (Funahashi et al., 2004). As stated above, local inactivation of either areas (8 or 9/46) results in localized deficits in the classic spatial working memory task. Combined with the recent study from our lab, the present results demonstrate that persistent activity across LPFC areas exhibit similar patterns of On/Off states, regardless of whether they are more involved in motor planning or non-effector-specific spatial working memory. Note however that it remains possible that nowhere in the brain is spatial delay activity completely independent of (motor) planning (e.g., Christophel et al., 2017; Ehrlich & Murray, 2022; Fine & Hayden, 2021). That remains an empirical question.

3. Not only did the recordings not encompass most of the PFC with presumed persistent activity, they certainly did not encompass other structures with persistent activity in spatial working memory. Most notably, Pat Goldman-Rakic showed many years ago (2000) that deactivation of PFC did not eliminate persistent activity during WM in parietal cortex, and deactivation of parietal cortex did not eliminate persistent activity during WM in PFC, suggesting that they were independent sources of persistent activity in working memory. Couldn't the parietal cortex hold the memory of the spatial location online during a PFC down state, and even restore the persistent activity during the up states, even if the PFC down states were global, not local?

As we describe above, first our recordings do encompass most of the LPFC with presumed persistent activity, particularly with the addition of data from the 3rd animal. Second, as we also detail above, it is already known that elimination of local activity within LPFC is sufficient to produce profound deficits in spatial working memory. This is even evident in the study referred to above (Chafee and Goldman-Rakic, 2000; Fig. R21). In short, loss of neuronal activity within LPFC is sufficient to destroy normal behavior. That is already known. In contrast, inactivation of parietal cortex results in comparatively mild deficits, even with larger inactivation volumes (e.g., Suzuki and Gottlieb, 2013).

Nonetheless, it is possible that populations in parietal cortex (or elsewhere) hold the memory via persistent spiking during the (spatially corresponding) Off state in LPFC, and that signal could be used to restore persistent activity during the On state. Indeed, our results do not address how On states might be restored. Our results do show that the patterns of functional connectivity preserve information about the memoranda during Off states, consistent with “activity-silent” models. But the mechanisms underlying the transition between the On and Off phases remains an open question. However, we do not draw strong conclusions about such mechanisms.

4. The idea that persistent activity in one structure might be eliminated but then restored due to persistent activity in another structure was specifically supported by the Svoboda lab (2016), at least for premotor activity. Deactivation of persistent activity in one hemisphere was restored by persistent activity in the other hemisphere. Furthermore, the Svoboda lab showed the importance of thalamo-cortical loops for persistent activity (2017).

To clarify, the reviewer is referring to evidence from mouse premotor cortex, not evidence from primate prefrontal cortex. Thus, the homology, or even analogy, is unclear. Given this, it is not surprising that in the first study cited (Li et al., 2016), unilateral silencing of premotor activity had very minimal effect on the behavior (motor preparation). This is in stark contrast to the primate LPFC where again localized inactivation results in robust impairments. Furthermore, in the primate, unilateral inactivation of LPFC eliminates persistent activity in most neurons in the thalamus (Alexander & Fuster, 1973). This discrepancy between mouse and primates presumably relates to the fact that medial prefrontal/premotor cortex in the mouse is not homologous to LPFC in the primate (e.g., Preuss, 1995)

Lastly, aside from the apparent lack of homology, the studies referred to are limited by issues that have left the question of persistent activity unresolved for some time, and which we address. Aside from a few exemplar rasters, the analyses focus on trial-averaged metrics of ‘persistence’. The fact that a trial-averaged spiking signal (similar to what we observe in our data) can be restored through interhemispheric effects following artificial inhibition does not reveal anything about the endogenous dynamics of neuronal populations on single trials and the presence or absence of a truly persistent signal.

5. The most intriguing evidence is that some pairwise cross-correlograms of cells are tuned to the cue during the delay. One thing I don’t understand is that the authors report that only 2% of the pairs recorded show tuned evoked responses to the cue during the cue period. Why are the responses of so few cells tuned to the cue? This seems different from what previous studies have reported, and it raises additional questions about whether this region is actually important for working memory.

We apologize for the confusion. In fact, 55% of neurons displayed a cue-specific evoked response, and we now state this explicitly in the main text (lines 325-326). The percentage

referred to by the reviewer reflects the proportion of the order n^2 pairs that, on average, are jointly selective for a specific 1 of the 8 possible cue locations. That quantity is useful for statistically assessing the relationship between CCGs and neural selectivity and does not describe the proportion of cells in the population that display tuning.

They also find that 1.5% of pairs showed significant CCGs to a given cue.

Indeed, we found this percentage reassuring, as it suggests that our CCG thresholding procedure was appropriately stringent. Note that the reviewer again refers to a percentage computed over pairs, not neurons, and conditioned on a particular cue. 81% of neurons participated in a cue-specific functional connection - we now clarify this in the text (lines 326).

They report that the percentage of cells tuned to the cue and participating in a CCG for the same cue during the delay is 2.5x greater than expected by chance. Fair enough, but it would be more useful to know what percentage of cells that respond selectively to a cue also participate in a CCG to the same cue during the delay. If I make a simple calculation myself, it seems like this is .075% of the cells showing this property. I accept that this is statistically significant, but it seems like an extremely small number of pairs to be the basis of claims about the basis for working memory.

The percentage of neurons that participate in jointly-selective and functionally connected pairs is in fact 10.4% (now clarified in lines 326-327) due to a similar source of confusion as in the previous two comments. While deciding whether or not a particular proportion of neurons should be considered large or small is an ill-defined exercise, the fact that, e.g., inhibitory cells comprise 10-20% of the neuronal population suggests that 10.4% may be a meaningful proportion.

6. Finally, I am not sure why the finding that some PFC pairs show cue-specific correlated activity during the delay means that their synapses must be facilitated. Couldn't the synapses be unmodified but the cells receive a subthreshold synaptic input from another source? This subthreshold input may influence the probability of joint spiking but not change firing rates. For example, maybe the thalamus or the parietal cortex targets pairs of cells in PFC that are tuned for a given cue location. I don't understand how this study localizes the source of plasticity to the synapses in the FEF.

The reviewer raises a couple of points worth clarifying.

First, to reemphasize, our recordings were only partly within the FEF, both in the original dataset, and in the current dataset which extends even further into 9/46.

Second, while our results provide the best evidence to date of a synaptic model of working memory, we do not make claims about localizing the "source of plasticity" within the LPFC. Instead, contrary to the dominant model of working memory, which specifically emphasizes persistent activity within the LPFC (e.g., Kandel et al., 2021), we show - for the first time - lapses

in mnemonic coding in LPFC on single trials. Furthermore, we show that those lapses are coupled with residual information about memoranda in the absence of a firing-rate code. The latter discovery is a key prediction of synaptic models (e.g., Mongillo et al., 2008), as we state explicitly in the discussion. While other mechanisms could underlie the type of cue-specific ensembles we observed, our results necessarily constrain any future comprehensive circuit/synaptic models of working memory.

References

- Alexander, G. E., & Fuster, J. M. (1973). Effects of cooling prefrontal cortex on cell firing in the nucleus medialis dorsalis. *Brain Research*, 61, 93–105. [https://doi.org/10.1016/0006-8993\(73\)90518-0](https://doi.org/10.1016/0006-8993(73)90518-0)
- Alonso, J. M., & Martinez, L. M. (1998). Functional connectivity between simple cells and complex cells in cat striate cortex. *Nature Neuroscience*, 1(5), 395–403. <https://doi.org/10.1038/1609>
- Ashwood, Z. C., Roy, N. A., Stone, I. R., Urai, A. E., Churchland, A. K., Pouget, A., & Pillow, J. W. (2022). Mice alternate between discrete strategies during perceptual decision-making. *Nature Neuroscience*, 25(2), 201–212. <https://doi.org/10.1038/s41593-021-01007-z>
- Barbosa, J., Stein, H., Martinez, R. L., Galan-Gadea, A., Li, S., Dalmau, J., Adam, K. C. S., Valls-Solé, J., Constantinidis, C., & Compte, A. (2020). Interplay between persistent activity and activity-silent dynamics in the prefrontal cortex underlies serial biases in working memory. *Nature Neuroscience*, 1–9. <https://doi.org/10.1038/s41593-020-0644-4>
- Bastos, A. M., Loonis, R., Kornblith, S., Lundqvist, M., & Miller, E. K. (2018). Laminar recordings in frontal cortex suggest distinct layers for maintenance and control of working memory. *Proceedings of the National Academy of Sciences*, 115(5), 1117–1122. <https://doi.org/10.1073/pnas.1710323115>
- Bruce, C. J., Goldberg, M. E., Bushnell, M. C., & Stanton, G. B. (1985). Primate frontal eye fields. II. Physiological and anatomical correlates of electrically evoked eye movements. *Journal of Neurophysiology*, 54(3), 714–734. <https://doi.org/10.1152/jn.1985.54.3.714>
- Busch, A., Roussy, M., Luna, R., Leavitt, M. L., Mofrad, M. H., Gulli, R. A., Corrigan, B., Mináč, J., Sachs, A. J., Palaniyappan, L., Muller, L., & Martinez-Trujillo, J. C. (2024). Neuronal activation sequences in lateral prefrontal cortex encode visuospatial working memory during virtual navigation. *Nature Communications*, 15(1), 4471. <https://doi.org/10.1038/s41467-024-48664-9>
- Chafee, M. V., & Goldman-Rakic, P. S. (2000). Inactivation of Parietal and Prefrontal Cortex Reveals Interdependence of Neural Activity During Memory-Guided Saccades. *Journal of Neurophysiology*, 83(3), 1550–1566. <https://doi.org/10.1152/jn.2000.83.3.1550>
- Christophel, T. B., Klink, P. C., Spitzer, B., Roelfsema, P. R., & Haynes, J.-D. (2017). The Distributed Nature of Working Memory. *Trends in Cognitive Sciences*, 21(2), 111–124. <https://doi.org/10.1016/j.tics.2016.12.007>
- Clark, K. L., Noudoost, B., & Moore, T. (2014). Persistent spatial information in the FEF during object-based short-term memory does not contribute to task performance. *Journal of Cognitive Neuroscience*, 26(6), 1292–1299. https://doi.org/10.1162/jocn_a_00599
- Constantinidis, C., Funahashi, S., Lee, D., Murray, J. D., Qi, X.-L., Wang, M., & Arnsten, A. F. T. (2018). Persistent Spiking Activity Underlies Working Memory. *Journal of Neuroscience*, 38(32), 7020–7028. <https://doi.org/10.1523/JNEUROSCI.2486-17.2018>

- Csicsvari, J., Hirase, H., Czurko, A., & Buzsáki, G. (1998). Reliability and State Dependence of Pyramidal Cell–Interneuron Synapses in the Hippocampus: An Ensemble Approach in the Behaving Rat. *Neuron*, 21(1), 179–189. [https://doi.org/10.1016/S0896-6273\(00\)80525-5](https://doi.org/10.1016/S0896-6273(00)80525-5)
- Donoho, D. L., & Johnstone, I. M. (1994). Ideal spatial adaptation by wavelet shrinkage. *Biometrika*, 81(3), 425–455. <https://doi.org/10.1093/biomet/81.3.425>
- Ehrlich, D. B., & Murray, J. D. (2022). Geometry of neural computation unifies working memory and planning. *Proceedings of the National Academy of Sciences*, 119(37), e2115610119. <https://doi.org/10.1073/pnas.2115610119>
- Fiebelkorn, I. C., Pinsk, M. A., & Kastner, S. (2018). A Dynamic Interplay within the Frontoparietal Network Underlies Rhythmic Spatial Attention. *Neuron*, 99(4), 842–853.e8. <https://doi.org/10.1016/j.neuron.2018.07.038>
- Fine, J. M., & Hayden, B. Y. (2021). The whole prefrontal cortex is premotor cortex. *Philosophical Transactions of the Royal Society B: Biological Sciences*, 377(1844), 20200524. <https://doi.org/10.1098/rstb.2020.0524>
- Funahashi, S., Bruce, C. J., & Goldman-Rakic, P. S. (1993). Dorsolateral prefrontal lesions and oculomotor delayed-response performance: Evidence for mnemonic “scotomas.” *The Journal of Neuroscience: The Official Journal of the Society for Neuroscience*, 13(4), 1479–1497. <https://doi.org/10.1523/JNEUROSCI.13-04-01479.1993>
- Funahashi, S., Chafee, M. V., & Goldman-Rakic, P. S. (1993). Prefrontal neuronal activity in rhesus monkeys performing a delayed anti-saccade task. *Nature*, 365(6448), 753–756. <https://doi.org/10.1038/365753a0>
- Funahashi, S., Takeda, K., & Watanabe, Y. (2004). Neural mechanisms of spatial working memory: Contributions of the dorsolateral prefrontal cortex and the thalamic mediodorsal nucleus. *Cognitive, Affective & Behavioral Neuroscience*, 4(4), 409–420. <https://doi.org/10.3758/CABN.4.4.409>
- González-Burgos, G., Barrionuevo, G., & Lewis, D. A. (2000). Horizontal synaptic connections in monkey prefrontal cortex: An in vitro electrophysiological study. *Cerebral Cortex (New York, N.Y.: 1991)*, 10(1), 82–92. <https://doi.org/10.1093/cercor/10.1.82>
- Jia, X., Tanabe, S., & Kohn, A. (2013). Gamma and the Coordination of Spiking Activity in Early Visual Cortex. *Neuron*, 77(4), 762–774. <https://doi.org/10.1016/j.neuron.2012.12.036>
- Jonikaitis, D., Noudoost, B., & Moore, T. (2023). Dissociating the Contributions of Frontal Eye Field Activity to Spatial Working Memory and Motor Preparation. *Journal of Neuroscience*, 43(50), 8681–8689. <https://doi.org/10.1523/JNEUROSCI.1071-23.2023>
- Kandel, E. R., Koester, J. D., Mack, S. H., & Siegelbaum, S. A. (2021). In *Principles of Neural Science*, 6e (1–Book, Section). McGraw Hill. accessbiomedicalscience.mhmedical.com/content.aspx?aid=1180370208
- Kobak, D., Brendel, W., Constantinidis, C., Feierstein, C. E., Kepecs, A., Mainen, Z. F., Qi, X.-L., Romo, R., Uchida, N., & Machens, C. K. (2016). Demixed principal component analysis of neural population data. *eLife*, 5, e10989. <https://doi.org/10.7554/eLife.10989>
- Li, N., Daie, K., Svoboda, K., & Druckmann, S. (2016). Robust neuronal dynamics in premotor cortex during motor planning. *Nature*, 532(7600), 459–464. <https://doi.org/10.1038/nature17643>
- Lundqvist, M., Rose, J., Herman, P., Brincat, S. L., Buschman, T. J., & Miller, E. K. (2016). Gamma and Beta Bursts Underlie Working Memory. *Neuron*, 90(1), 152–164. <https://doi.org/10.1016/j.neuron.2016.02.028>
- Mongillo, G., Barak, O., & Tsodyks, M. (2008). Synaptic Theory of Working Memory. *Science*, 319(5869), 1543–1546. <https://doi.org/10.1126/science.1150769>
- Murray, J. D., Bernacchia, A., Roy, N. A., Constantinidis, C., Romo, R., & Wang, X.-J. (2017). Stable population coding for working memory coexists with heterogeneous neural

- dynamics in prefrontal cortex. *Proceedings of the National Academy of Sciences*, 114(2), 394–399. <https://doi.org/10.1073/pnas.1619449114>
- Petrides, M. (2005). Lateral prefrontal cortex: Architectonic and functional organization. *Philosophical Transactions of the Royal Society B: Biological Sciences*, 360(1456), 781–795. <https://doi.org/10.1098/rstb.2005.1631>
- Preuss, T. M. (1995). Do rats have prefrontal cortex? The rose-woolsey-akert program reconsidered. *Journal of Cognitive Neuroscience*, 7(1), 1–24. <https://doi.org/10.1162/jocn.1995.7.1.1>
- Quiroga, R. Q., Nadasdy, Z., & Ben-Shaul, Y. (2004). Unsupervised spike detection and sorting with wavelets and superparamagnetic clustering. *Neural Computation*, 16(8), 1661–1687. <https://doi.org/10.1162/089976604774201631>
- Rainer, G., Asaad, W. F., & Miller, E. K. (1998). Memory fields of neurons in the primate prefrontal cortex. *Proceedings of the National Academy of Sciences of the United States of America*, 95(25), 15008–15013. <https://doi.org/10.1073/pnas.95.25.15008>
- Sawaguchi, T., & Iba, M. (2001). Prefrontal cortical representation of visuospatial working memory in monkeys examined by local inactivation with muscimol. *Journal of Neurophysiology*, 86(4), 2041–2053. <https://doi.org/10.1152/jn.2001.86.4.2041>
- Senzai, Y., Fernandez-Ruiz, A., & Buzsáki, G. (2019). Layer-Specific Physiological Features and Interlaminar Interactions in the Primary Visual Cortex of the Mouse. *Neuron*, 101(3), 500–513.e5. <https://doi.org/10.1016/j.neuron.2018.12.009>
- Siegle, J. H., Jia, X., Durand, S., Gale, S., Bennett, C., Graddis, N., Heller, G., Ramirez, T. K., Choi, H., Luviano, J. A., Groblewski, P. A., Ahmed, R., Arkhipov, A., Bernard, A., Billeh, Y. N., Brown, D., Buice, M. A., Cain, N., Caldejon, S., ... Koch, C. (2021). Survey of spiking in the mouse visual system reveals functional hierarchy. *Nature*, 592(7852), 86–92. <https://doi.org/10.1038/s41586-020-03171-x>
- Smith, M. A., & Kohn, A. (2008). Spatial and Temporal Scales of Neuronal Correlation in Primary Visual Cortex. *Journal of Neuroscience*, 28(48), 12591–12603. <https://doi.org/10.1523/JNEUROSCI.2929-08.2008>
- Suzuki, M., & Gottlieb, J. (2013). Distinct neural mechanisms of distractor suppression in the frontal and parietal lobe. *Nature Neuroscience*, 16(1), 98–104. <https://doi.org/10.1038/nn.3282>
- Tao, W., & Libedinsky, C. (2024). *Evidence of Activity-Silent Working Memory in Prefrontal Cortex* (p. 2024.06.03.597259). bioRxiv. <https://doi.org/10.1101/2024.06.03.597259>
- Trepka, E. B., Zhu, S., Xia, R., Chen, X., & Moore, T. (2022). Functional interactions among neurons within single columns of macaque V1. *eLife*, 11, e79322. <https://doi.org/10.7554/eLife.79322>
- Verkuilen, J., & Smithson, M. (2012). Mixed and mixture regression models for continuous bounded responses using the beta distribution. *Journal of Educational and Behavioral Statistics*, 37(1), 82–113. <https://doi.org/10.3102/1076998610396895>

Reviewer Reports on the First Revision:

Referees' comments:

Referee #1 (Remarks to the Author):

I have read the authors' response and revised manuscript. My questions have been adequately addressed, and the data and analyses that have been added significantly improve the manuscript's impact. Overall, I think this is an important and timely study that will be of significant interest to a broad range of neuroscientists, particularly those who study the neurophysiological basis of cognition.

Referee #2 (Remarks to the Author):

The authors have successfully addressed all my (potential) concerns. This is a solid revision to an important manuscript, with the key results now significantly reinforced. My previous assessment stands that these results will be of high interest to the broader field.

Eelke Spaak

Referee #3 (Remarks to the Author):

The authors have collected more data and significantly revised the manuscript, resulting in more convincing conclusion about on an off states in working memory, and the role of potentiated synapses. The results are important and will be of interest to many people. I have a small number of additional questions that are mainly discussion points.

1. Although the CCGs between cue-selective neurons during the delay period are evidence for potentiated synapses holding the cue memory during the off periods, there is something about the explanation that does not make sense to me. The authors find CCGs between the cue-selective cells during the cue period, and also find them during the off periods. But why are the CCGs not significant during the on periods in the delay? The authors argue that synapses undergo STDP during the cue, which sensitize their connections throughout the delay, including the off periods, but if so, then why would these potentiated connections (evidenced by CCGs) not persist during the on periods? If they are gone during the on periods, how could they re-emerge during the next off period? This at least deserves some discussion.
2. Another puzzling finding is that if an off period occurs at the end of the working memory interval, just before the delayed behavioral response, behavioral performance is worse than if an on period occurred just before the delayed behavioral response. This finding is barely mentioned in the Results and appears in a supplementary figure, but it seems like it has an important bearing on the significance of the on versus off periods. Most of all, it suggests that it is the persistent spiking activity that is more important for behavioral performance in working memory than the potentiated synapses. Another puzzle is that behavioral performance drops in all three monkeys when the off period occurs before the behavioral response, but the drop in performance is relatively small in two

of the animals (Fig S2B). How to explain this? One possibility is that the cortex recorded around one electrode is not representative of all of LPFC, and off states at one location are accompanied by on states at other locations. This is related to my next point.

3. I noted in my previous review that the recordings were confined to a small part of LPFC, so it is hard to know if other regions showed persistent activity when the cortex around the electrode was in an off state. The authors expanded the area of the recordings, which is good, but that really doesn't address the concern. The authors also responded that deactivation studies have shown that deactivating any part of PFC severely impairs working memory. However, I think the authors misunderstood my point. It could easily be that all parts of LPFC contribute (or other brain areas such as parietal cortex) to working memory on average, but the off states in one local region at one moment in time are accompanied by on states of persistent activity in other regions. In that case, persistent activity (in the on states) could be the most important mechanism for working memory. Indeed, the behavioral results I mention in point 2 are consistent with this idea. This question could only be answered by simultaneous recordings spread apart over PFC. It would be unreasonable to expect the authors to have such data, but it does seem like this possibility is worth some discussion.